# A nanobody-based molecular toolkit provides new mechanistic insight into clathrin-coat initiation

Linton M Traub*

Department of Cell Biology, School of Medicine, University of Pittsburgh, Pittsburgh, United States

**Abstract** Besides AP-2 and clathrin triskelia, clathrin coat inception depends on a group of early-arriving proteins including Fcho1/2 and Eps15/R. Using genome-edited cells, we described the role of the unstructured Fcho linker in stable AP-2 membrane deposition. Here, expanding this strategy in combination with a new set of llama nanobodies against EPS15 shows an FCHO1/2–EPS15/R partnership plays a decisive role in coat initiation. A nanobody containing an Asn-Pro-Phe peptide within the complementarity-determining region 3 loop is a function-blocking pseudoligand for tandem EPS15/R EH domains. Yet, in living cells, EH domains gathered at clathrin-coated structures are poorly accessible, indicating residence by endogenous NPF-bearing partners. Forcibly sequestering cytosolic EPS15 in genome-edited cells with nanobodies tethered to early endosomes or mitochondria changes the subcellular location and availability of EPS15. This combined approach has strong effects on clathrin coat structure and function by dictating the stability of AP-2 assemblies at the plasma membrane.
DOI: https://doi.org/10.7554/eLife.41768.001

## Introduction

Clathrin-mediated endocytosis is an evolutionarily ancient mode of directed transport of selected macromolecules into the eukaryotic cell interior. The process depends on the deposition of an ordered precession of cytosolic coat components at a small, flat, demarcated zone of the plasma membrane destined to transform morphologically into a spherical clathrin-coated vesicle (*Avinoam et al., 2015*; *Bucher et al., 2018*). Generally, the local assembly of the organized coat coordinates the process of membrane deformation into an invaginated bud with the recognition and capture of transmembrane cargo (*Kaksonen and Roux, 2018*; *Mettlen et al., 2018*). To properly build the coat, a succession of discrete coat proteins display different kinetics of arrival and departure (*Taylor et al., 2011*), and can be differentially positioned within the assembling polyhedral clathrin lattice (*Sochacki et al., 2017*; *Ma et al., 2016*) that is emblematic of this type of intracellular transport vesicle coat. The so-called pioneer proteins (*Ma et al., 2016*; *Traub and Wendland, 2010*), which arrive first to define and arrange the incipient bud site, include epidermal growth factor receptor pathway substrate 15 (Eps15), the core heterotetrameric clathrin adaptor AP-2, and FCH/F-BAR domain only protein 1 (Fcho1) and Fcho2 (*Taylor et al., 2011*). Eps15 is a fibrous (>30 nm) protein oligomer (*Tebar et al., 1997*; *Cupers et al., 1997*), originally identified as a tyrosine phosphorylated substrate of the activated EGF receptor, hence the designation Eps15 (*Fazioli et al., 1993*). Yet even in the absence of EGF receptor activation, Eps15 localizes to the vast majority of cell surface clathrin-coated structures at steady state (*van Delft et al., 1997a*; *Benmerah et al., 1999*; *Tebar et al., 1996*), is positioned predominantly at the margins of assembling lattices (*Sochacki et al., 2017*; *Tebar et al., 1996*; *Henne et al., 2010*; *Edeling et al., 2006*), and is among the first assembled coat proteins encountered by an activated G protein-coupled

*For correspondence:
traub@pitt.edu

Competing interests: The author declares that no competing interests exist.

receptor (GPCR) moving laterally in the plane of the plasma membrane into a preassembled clathrin-coated region (*Paek et al., 2017*).

A large multi-domain protein, Eps15 binds physically to numerous other coat components, including AP-2 (*Edeling et al., 2006*; *Benmerah et al., 1996*; *Traub et al., 1999*; *Schmid et al., 2006*), Fcho1/2 (*Henne et al., 2010*; *Reider et al., 2009*; *Uezu et al., 2011*; *Umasankar et al., 2012*), Disabled-2 (Dab2) (*Teckchandani et al., 2012*), epsin 1 (*Chen et al., 1998*; *McPherson et al., 1998*), HIV-1 Rev-binding protein (Hrb) (*Doria et al., 1999*; *Whitehead et al., 1999*; *Yamabhai et al., 1998*), Numb (*Santolini et al., 2000*; *Evergren et al., 2018*), stonin 2 (*Rumpf et al., 2008*), and synaptojanin 1 (*McPherson et al., 1998*; *Haffner et al., 1997*), among other proteins including ubiquitin (*Hofmann and Falquet, 2001*). Because of this multiplicity of protein–protein contacts, residence of Eps15 at clathrin bud sites is not solely dependent on AP-2 (*van Delft et al., 1997a*). While much is known about the biology of Eps15 (*Teckchandani et al., 2012*; *van Bergen En Henegouwen, 2009*; *Milesi et al., 2019*), several aspects of its operation still remain opaque, including how the multiple synchronous protein–protein interactions might be prioritized and regulated, how asymmetric partitioning within a clathrin-coated structure is achieved, and how post-translational modifications, including phosphotyrosine addition (*Torrisi et al., 1999*) and ubiquitination (*Polo et al., 2002*; *van Delft et al., 1997b*; *Francavilla et al., 2016*), impact Eps15 operation and stability.

Numerous complimentary experimental approaches are available in contemporary cell biology to probe the structure, location and function of individual gene products. Most notably, genome-editing technology provides an unparalleled ability to disrupt or modify gene products in a cellular setting (*Smith et al., 2017*; *Housden et al., 2017*). Transcription activator-like effector nuclease (TALEN)-mediated genome editing is used here to silence the EPS15R-encoding *EPS15L1* locus in a HeLa cell line that also lacks the expression of the pioneer proteins FCHO1 and FCHO2 (*Umasankar et al., 2014*). Other technically useful current tools for biochemical and cellular analyses are single chain nanobodies (Nbs) derived from *Camelidae* species (*Beghein and Gettemans, 2017*; *Wang et al., 2016a*). Since the variable heavy-chain domain from heavy chain antibodies (V$_H$H) encoded by Nbs is only a single, stably folded, compact chain of ~13 kDa, they are easy to subclone, express and transfect (*Moutel et al., 2016*; *Dmitriev et al., 2016*). They are further versatile as the smallest, autonomous native antigen-binding fold in that ectopically expressed monomeric V$_H$H fragments often remain operational in the reduced cytosolic environment (*Moutel et al., 2016*; *Pleiner et al., 2015*; *Schenck et al., 2017*). Here, a set of anti-Eps15 Nbs is characterized biochemically and an assortment of Nb-based fusion proteins for cell-based analysis evaluated.

## Results

### Identification of anti-EPS15 EH domain Nbs

A phage-based immune llama (*Lama glama*) V$_H$H library, raised against residues 1–314 of the *Homo sapiens* (*Hs*) EPS15 protein from chromosome 1 (*Wong et al., 1994*), was generated and screened. As Nbs preferentially recognize folded conformational antigens over linear or peptide regions, the immunizing antigen corresponds to three consecutively-arrayed Ca$^{2+}$-binding, EF-hand-like Eps15 homology (EH) domains (designated EH1-EH3) (*Figure 1A*), where the individual tandem EH domains are about 32% identical and 50% similar to one another. The divergence between EH domain sequences is paralleled by differences in selectivity for endogenous tripeptide-based Asn-Pro-Phe (NPF) short linear motif (SLiM) (*Van Roey et al., 2014*) -containing binding partners (*Rumpf et al., 2008*; *Paoluzi et al., 1998*; *Confalonieri and Di Fiore, 2002*). Overall, EPS15 residues 1–314 are ~60% identical and 77% similar to resides 1–365 of EPS15R, a closely related paralogous protein encoded by chromosome 19 (*Wong et al., 1995*) (*Figure 1A*).

Comparative sequence analysis of the seven ELISA-positive V$_H$H clones reveals three discrete families (*Figure 1B*), albeit because of an identical hypervariable complementarity-determining region 3 (CDR3) (*Figure 1C*), family 2 and 3 might be derived from the same B cell lineage that diverge due to somatic-mutation-driven affinity maturation and/or PCR amplification errors. There are 18 amino acid differences between Nb E_142 and E_180, but only six of the changes are within CDR1 and CDR2. This sequence variation between family 2 and 3 is curious because the CDR3 loop is typically the longest, most divergent in amino acid composition, conformationally variable, and important for antigen recognition (*Mitchell and Colwell, 2018*; *McMahon et al., 2018*). The three

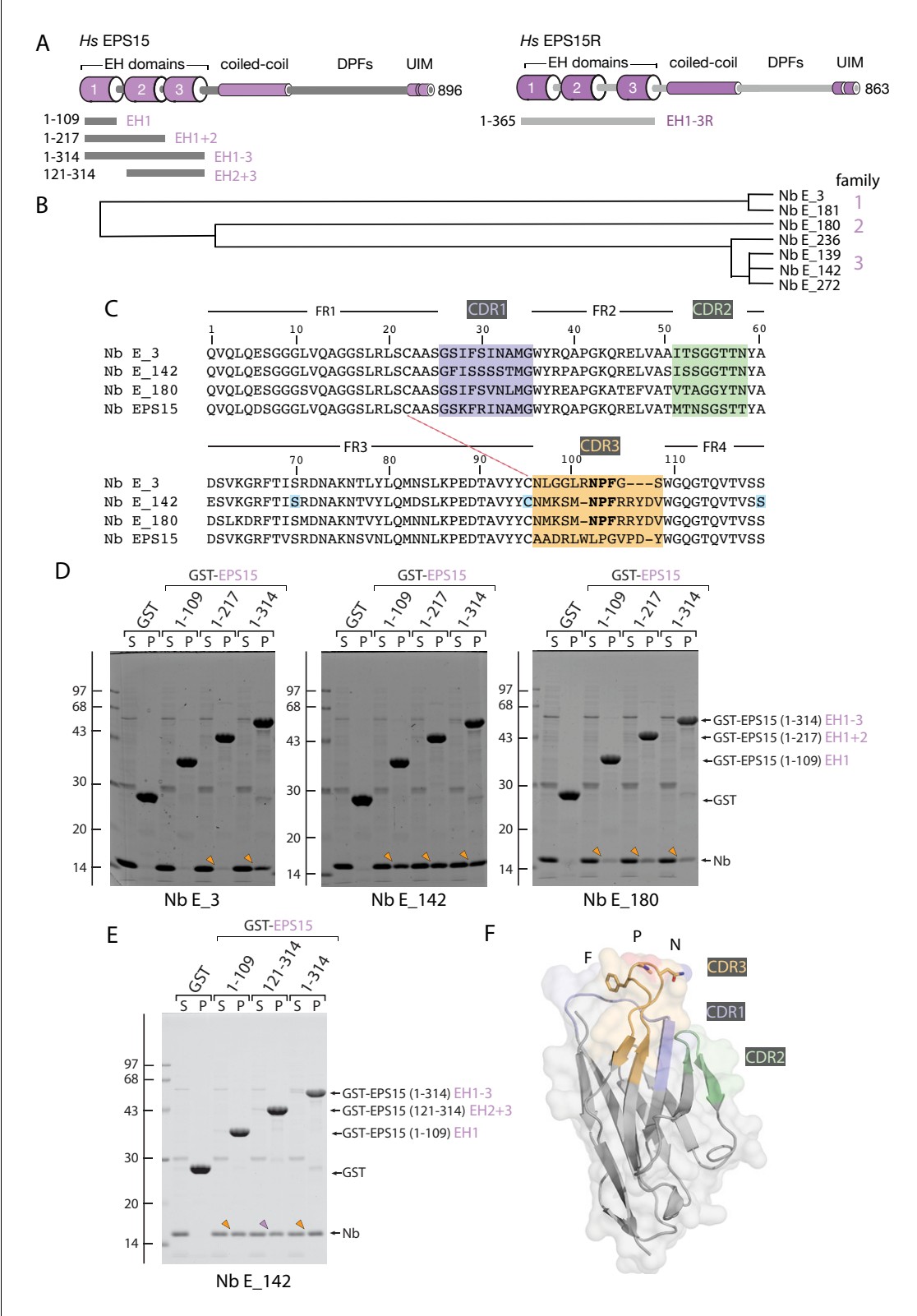

**Figure 1.** Anti-EPS15 Nb identification. (**A**) Schematic diagrams of the overall architecture and structural features of *H. sapiens* (*Hs*) EPS15 and EPS15R proteins. The relative location of various GST-fusion proteins used is indicated. DPF: Asp-Pro-Phe SLiM repeat containing domain, UIM: ubiquitin-interacting motif. (**B**) Dendrogram of amino-acid sequence relationship between the seven V$_H$H anti-EPS15 Nbs selected by biopanning and ELISA screening. Some intrafamily variability may have arisen from RT-PCR and/or amplification PCR-based DNA polymerase errors. (**C**) Amino acid alignment.
*Figure 1 continued on next page*

Figure 1 continued

Framework regions (FR) 1–4 and interposed, color-coded CDR1-3 stretches are indicated. The pair of cysteine residues involved in disulfide formation is indicated (dashed red line), while the three residues mutated in the conditionally-destabilized (*Tang et al., 2016*) desNb E_142 protein are highlighted (blue). (D) Pull-down assay of soluble anti-EPS15 Nb in *E. coli* periplasmic lysates using 50 µg GST, GST-EPS15 (1-109 , 1-217) or (1-314). Analysis of supernatant (S) and pellet (P) fractions after incubation of Sepharose-bead-immobilized GST fusion with *E. coli* periplasmic extract containing the indicated Nb. Coomassie-stained gels shown, with the position of the molecular mass standards (in kDa) indicated. Bound Nb recovered in the pellet fraction is indicated (arrowheads). (E) Binding of Nb E_142 to GST-EPS15 (1-134) and (121-314) lacking the EH1 domain as in D. (F) Combined ribbon and molecular surface representation of a computationally-threaded structure of Nb E_142 modeled by Phyre2 server (*Kelley et al., 2015*). The locations of the CDR1-3 on the folded $V_H H$ domain model are indicated with coloring as in C, while the NPF SLiM in CDR3 is shown in stick representation and single letter amino acid code.

DOI: https://doi.org/10.7554/eLife.41768.002

unique Nb sequences selected for detailed further analysis (one from each family; designated E_3, E_142 and E_180) are dissimilar to that of a previously reported anti-EPS15 Nb isolated against EPS15 EH1-3 domains from a naïve llama library (*Regan-Klapisz et al., 2005*) (*Figure 1C*).

In in vitro pull-down assays, a direct physical interaction between each of the chosen Nbs with the EPS15 N-terminal EH domain antigen is seen (*Figure 1D*). Nb E_3 binds to GST-EPS15 EH1-3 (residues 1–314), but poorly to GST fused in-frame to either domain EH1 alone (residues 1–109) or EH1 + 2 (residues 1–217). Not unexpectedly, Nb E_142 and E_180 show similar binding selectivity, in accordance with the shared CDR3 sequences of these two Nb clones. However, Nb E_142 clearly shows a higher apparent affinity, and interacts with all three EH domain proteins, EH1, EH1 + 2 and EH1-3 (*Figure 1D*). One interpretation of the data is that Nb E_3 recognizes the EH3 domain while Nb E_142 (and E_180) binds to the EH1 domain. Yet Nb E_3 does show appreciable binding to GST-EPS15 EH1 + 2, and E_142 binds to GST-EPS15 (1-314) at perhaps suprastoichiometric levels, and does not require EH1. A robust interaction of Nb E_142 with GST-EPS15 EH2 + 3 (residues 121–314) occurs in addition to binding to the EH1 domain alone (*Figure 1F*); this interaction with a GST-fusion lacking the EH1 domain confirms specific binding of Nb E_142 to EH domains other than EH1.

Intriguingly, Nb E_3, E_142 and E_180, as well as the other four related Nb clones, all show the remarkable presence of an immunodominant Asn-Pro-Phe (NPF) tripeptide within the CDR3 region (*Figure 1C*). A BLAST analysis of available Nb sequences using the Nb E_142 sequence as the search query fails to identify a single NPF sequence within the CDR3 hypervariable loop of any currently characterized $V_H H$ clone. Only a series of selected anti-*Tulip virus X* Nbs derived from a single alpaca contain either an NPW or NPV tripeptide sequence within the CDR3 loop (*Beekwilder et al., 2008*). As the majority of EH domain partners (the so-called EH network) deploy some variation of the NPF SLiM (*Paoluzi et al., 1998*; *Confalonieri and Di Fiore, 2002*) to bind to an EH domain, this strongly suggests that the newly selected anti-EPS15 EH domain Nbs bind, in part, to the functional interaction surface of the EH domain (*de Beer et al., 1998*). In agreement, the previously described anti-EPS15 Nb does not contain an NPF within CDR3 (*Figure 1C*), does not bind to the native EPS15 protein efficiently (data not shown), and was originally selected to detect (denatured) EPS15 by immunoblotting (*Regan-Klapisz et al., 2005*).

Direct evidence for the participation of the CDR3 NPF tripeptide in EH domain binding comes from site-directed mutagenesis. Compared with the native $V_H H$, a Nb E_142 (NPF→APA) replacement eliminates completely the interaction with GST-EPS15 (1-314) (*Figure 2A*). Because of the minimal nature of the NPF SLiM, with a single hydrophobic anchor (Phe) residue, individual internal NPF motifs generally bind to EH domains rather weakly, with an equilibrium dissociation constant ($K_D$) of >100 µM (*Yamabhai et al., 1998*; *de Beer et al., 1998*). Cyclized (stapled) NPF peptides bind with higher apparent affinity (~10 fold, *de Beer et al., 2000*) due to stabilizing conformational rigidity (*Yamabhai et al., 1998*), since the NPF adopts a type I β-tight turn when engaging an EH domain (*de Beer et al., 2000*). It seems likely that the structural constraints of the overall Nb immunoglobulin (Ig) superfamily fold and the CDR3 loop similarly increase the apparent affinity of the Nbs for the EH domain, although contributions of CDR1 and CDR2 are also likely. The differing relative positioning of the NPF in the CDR3 loop may be responsible for the lower apparent affinity of the Nb E_3 clone and/or more involvement of the CDR1 and -2 loops in engagement of the EH3 domain.

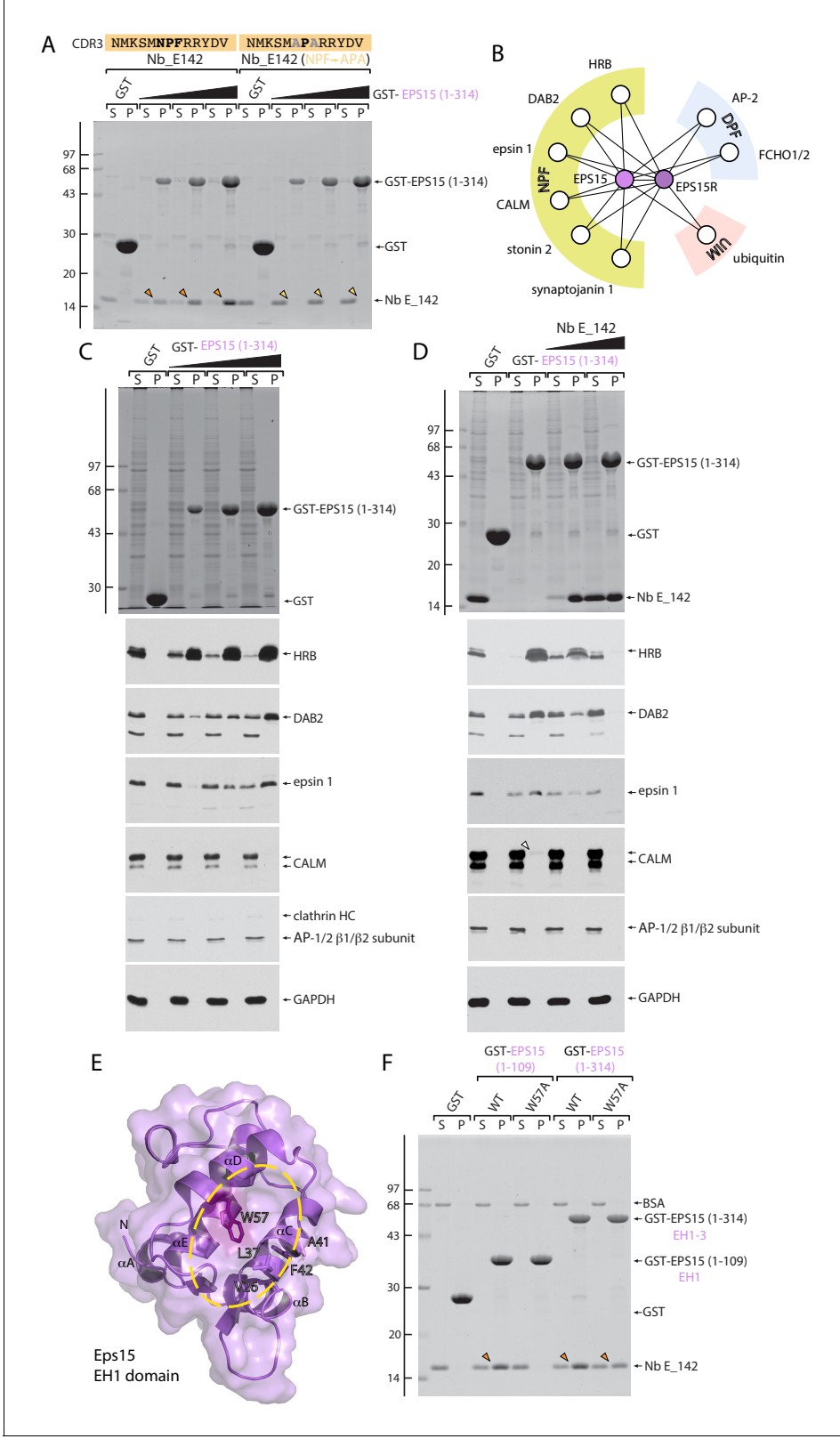

**Figure 2.** EH domain function-blocking Nb E_142. (**A**) Pull-down binding assay of Nb E_142 and an NPF→APA E_142 mutant expressed in *E. coli* periplasmic extracts to either GST or increasing amounts of GST-EPS15 (1-314). Typical Coomassie-stained gel shown with the position of the molecular mass standards (in kDa) indicated on the left. (**B**) Schematic depiction of EPS15 and EPS15R interaction partners grouped by peptide SLiM or protein interaction type. (**C**) Biochemical pull-down assay of NPF-SLiM-containing partners within HeLa whole cell lysate to either immobilized GST or

*Figure 2 continued on next page*

*Figure 2 continued*

increasing amounts of GST-EPS15 (1-314) EH1-3 domain. After fractionation by SDS-PAGE, gels were either stained with Coomassie blue or transferred to nitrocellulose and immunoblotted with the indicated antibodies. (D) Inhibition of NPF-based partner protein interactions with 200 µg immobilized GST-EPS15 (1-314) by equimolar (45 µg) or 5-fold (225 µg) molar excess of added Nb E_142. (E) Combined ribbon and molecular surface representation of the *M. musculus* Eps15 EH1 domain (PDB ID: 1QJT) with location of W57 highlighted. Residues within the interaction surface (yellow oval) that undergo substantial chemical shift upon NPF ligand binding (*Whitehead et al., 1999*) are indicated. (F) Elimination of Nb binding by EH1 domain interaction site disrupting W57A mutation. Coomassie-stained gel of a pull-down assay using 50 µg of the indicated GST-fusion proteins with purified Nb E_142 protein. The location of the BSA carrier protein in the supernatant fractions is indicated, and bound Nb recovered in pellet fractions is indicated (arrowheads).

DOI: https://doi.org/10.7554/eLife.41768.003

## Characterization of a function-blocking Nb

As a protein–protein interaction module, in biochemical pull-down assays, the immobilized GST-EPS15 (1-314) EH1-3 array interacts physically with several NPF-harboring cytosolic partners from HeLa cell lysate, in a dose-dependent manner (*Figure 2B and C*). The apparent affinity correlates roughly with the number of NPF SLiMs (HRB: 5; DAB2: 5; epsin 1: 3; c̲lathrin a̲ssembly l̲ymphoid m̲yeloid leukemia protein (CALM): 2) and not with the presence an extreme C-terminal NPF repeat (HRB, DAB2 and epsin 1) (*Yamabhai et al., 1998*), with HRB being the most avid binding partner (*Doria et al., 1999*; *Whitehead et al., 1999*; *Salcini et al., 1997*). Both major isoforms of CALM contain two NPFs, one of which is near the C-terminus but not as close as occurs in either DAB2 or HRB. However, only very limited binding of cytosolic CALM to the immobilized EH domain array is evident (*Figure 2D*). In this assay, the engagement of cytosolic NPF partners by GST-EPS15 EH1-3 is affected by highest apparent affinity Nb E_142 protein; thus the $V_H$H clone is function blocking. Supplementing HeLa cell lysate with Nb E_142 at an equimolar amount relative to the GST-EPS15 (1-314) causes a substantial loss of DAB2 and epsin 1 from the bound pellet fraction and incomplete inhibition of the HRB association. A fivefold molar excess of Nb E_142 produces near complete obstruction of HRB, DAB2 and epsin 1 interactions, concomitant with recovery of the bound $V_H$H domain with the GST-EPS15 fusion in the pellet fraction (*Figure 2D*). In addition, a W57A substitution in the EPS15 EH1 domain (*Whitehead et al., 1999*), which involves the absolutely phylogenetically conserved Trp residue in all EH domains (*Confalonieri and Di Fiore, 2002*; *de Beer et al., 1998*; *de Beer et al., 2000*) and disrupts the canonical NPF interaction surface between α-helices αC and αD (*Figure 2E*) (*Rumpf et al., 2008*), abolishes the association of Nb E_142 with GST-EPS15 (1-109) (*Figure 2F*). Yet the same W57A mutation has a comparatively modest effect on binding in the context of the larger GST-EPS15 (1-314) fusion protein (*Figure 2F*). Collectively, these data clearly indicate that Nb E_142 engages EH domains as an NPF-mimicking pseudoligand that contacts a conserved interaction site upon the canonical EH domain.

If the Nbs bind to EH domains primarily utilizing an NPF-motif-mediated interaction, then reactivity with the related EPS15R EH domains is highly probable as the EPS15 and EPS15R EH domain regions are ~80% similar. In EPS15R, both the EH1R and EH3R domains favor NPFR motifs (*Paoluzi et al., 1998*) and the position +4 Arg participates directly in binding to the EPS15 EH2 domain (*de Beer et al., 2000*). Strikingly, two of three of the Nb families identified display NPFR peptide sequences in CDR3 (*Figure 1C*). In line with the biochemical differences noted for GST-EPS15 (1-314) engagement, both Nb E_3 and E_142 cross-react with a GST-EPS15R (1-365) fusion, while the Nb E_142 (NPF→APA) CDR3 mutant does not (*Figure 3A*). Nb E_3 binding to EPS15R is comparatively weak, however. In sum, the biochemical data indicate that Nb E_142 is an orthosteric, function-blocking pseudoequivalent NPF ligand for the Eps15/R protein EH domains.

The EH domains of intersectin 1 similarly bind to NPF motifs (*Yamabhai et al., 1998*), and the two N-terminal *Hs* intersectin 1 EH domains are about 43% identical to one another while the EH1-2 array (residues 1–312) (*Snetkov et al., 2016*) is roughly 39% identical to EPS15 EH1-3 (residues 1–314) and 36% identical to EPS15R EH1R-3R (residues 1–365). Yet in the same binding assay, Nb E_142 does not interact appreciably with GST-intersectin 1 (1-312) although the unmistakable Nb binding to both GST-EPS15 (1–109/EH1) and GST-EPS15 (1–217/EH1 + 2) is apparent (*Figure 3B*). This indicates importantly that additional non-NPF side chains must contribute to the selective binding of Nb E_142 to the EPS15/R EH domains, and likely accounts for the markedly higher apparent affinity of this Nb over the native NPF ligand for EPS15/R EH domains. Also, since none of the anti-

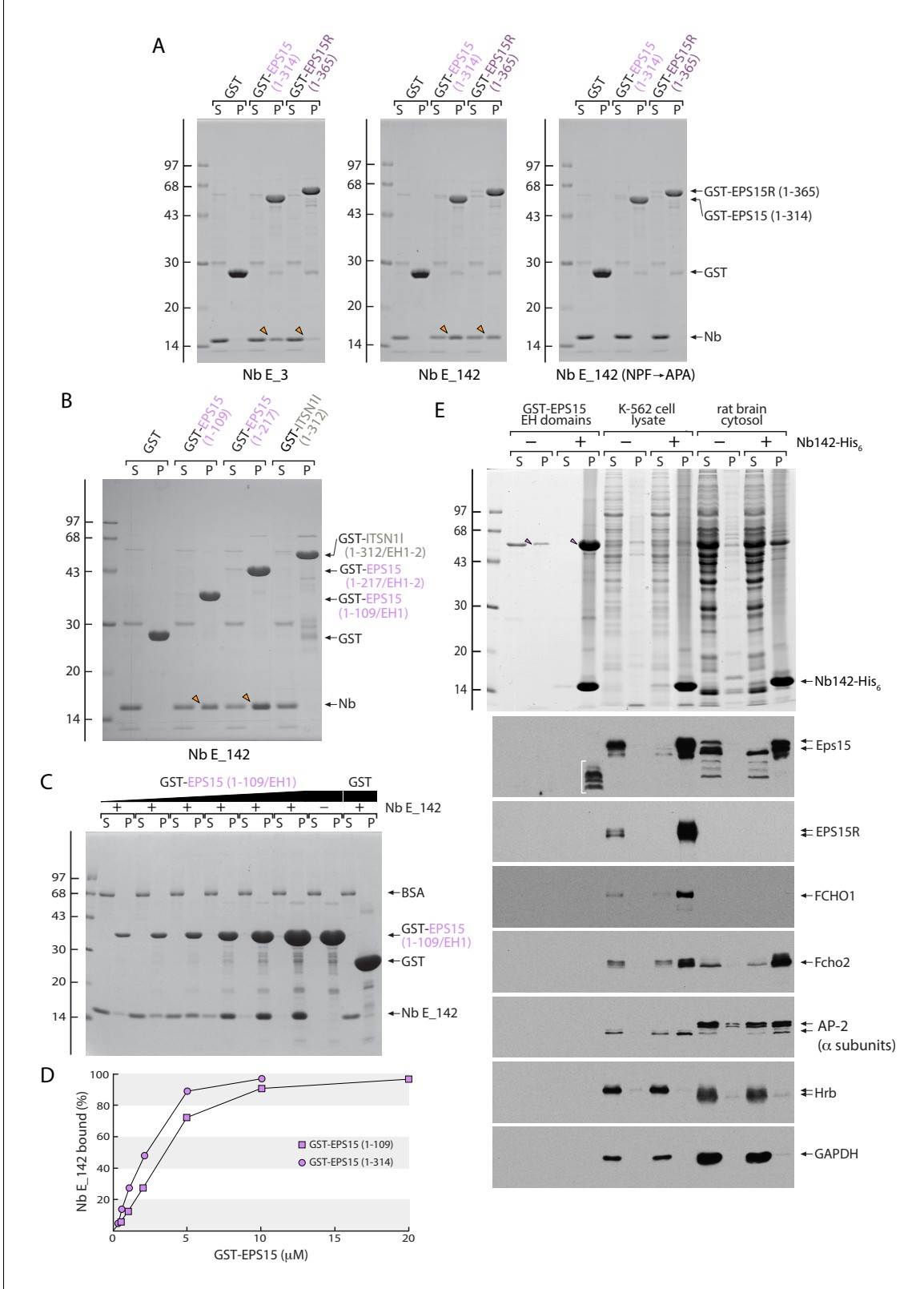

**Figure 3.** Direct interaction of Nb E_142 with the EPS15/R but not intersectin 1 EH domains. (**A**) Analysis of supernatant (S) and pellet (P) fractions after incubation of 50 µg Sepharose-bead-immobilized GST, GST-EPS15 (1-314) or GST-EPS15R (1-365) with periplasmic extracts containing either Nb E_3, Nb E_142 or Nb E_142 (NPF→APA) as indicated. Coomassie-stained gels shown, with the position of the molecular mass standards (in kDa) indicated on the left, and bound Nb (arrowheads) shown. (**B**) Analysis of supernatant (S) and pellet (P) fractions after incubation of 50 µg Sepharose-bead-

*Figure 3 continued on next page*

*Figure 3 continued*

immobilized GST, GST-EPS15 (1–109/EH1) or GST-EPS15 (1–217/EH1-2) and GST-intersectin 1 l (ITSN1l) (1–312/EH1-2) with periplasmic extracts containing either Nb E_3, Nb E_142 as indicated. Coomassie-stained gel shown, with the position of the molecular mass standards (in kDa) indicated on the left, and bound Nb (arrowheads) shown. (C) Analysis of supernatant (S) and pellet (P) fractions after incubation of 20 µM Sepharose-bead-immobilized GST or 0.5–20 µM GST-EPS15 (1–109/EH1) with 2 µM purified Nb E_142 in the presence of 100 µg/ml BSA as indicated. Coomassie-stained gel shown, with the position of the molecular mass standards (in kDa) indicated on the left. (D) Plots of the mean of three determinations that differed by less than 10% shown for GST-EPS15 (1-109) (squares) or GST-EPS15 (1-314) (circles) titrations. Estimation of $K_D$ from the quantitative analysis of the supernatant fractions yields ~3 µM for a single EH domain (1-109) and ~2 µM for the triple EH domain array (1-314). (E) Analysis of supernatant and pellet fractions after immunoprecipitation of purified GST-EPS15 (1-314) (arrowheads) or native Eps15 in K-562 cell lysate or rat brain cytosol. After fractionation by SDS-PAGE, gels were either stained with Coomassie blue or transferred to nitrocellulose and immunoblotted with the indicated antibodies. Non-specific rabbit anti-EPS15 cross-reactive material in the pellet fraction from the GST-EPS15 (1-314) incubation is identified (bracket).

DOI: https://doi.org/10.7554/eLife.41768.004

The following figure supplement is available for figure 3:

**Figure supplement 1.** Quantitative titration of Nb E_142 with immobilized GST-EPS15 (1-314).

DOI: https://doi.org/10.7554/eLife.41768.005

EPS15 Nbs display acidic residues trailing the NPF in CDR3, it makes it improbable that Nb E_142 binds to other proteins with C-terminal EH domains, like EHD1 (*Kieken et al., 2010*).

Biochemical titrations better gauge the affinity of Nb E_142 for EPS15 EH domains. Using 2 µM Nb E_142 and immobilized GST-EPS15 (1–109/EH1) varied from 0.5 to 20 µM, saturation occurs at around 10 µM and half-maximal binding occurs at ~3 µM (*Figure 3C–D*). With the larger GST-EPS15 (1–314/EH1-3) fusion, half-maximal binding is also seen at ~2 µM (*Figure 3D* and *Figure 3—figure supplement 1*). That the apparent $K_D$ for Nb E_142 is roughly similar for the two fusion proteins indicates that binding to each EH domain is independent and of similar affinity. Thus Nb E_142 binds to the EH domain with low micromolar affinity at physiological ionic strength; these results demonstrate that the $K_D$ of Nb E_142 for an EH domain is considerably better than a single endogenous NPF peptide (*Yamabhai et al., 1998*; *de Beer et al., 1998*).

Nb E_142 is an immunoprecipitation-grade reagent. In addition to the binary interaction with recombinant GST-EPS15 (1-314) antigen (*Figure 3E*), purified His₆-tagged Nb E_142 bound to Ni-NTA-agarose efficiently immunoprecipitates Eps15 from both K-562 erythroleukemia cell (*Lozzio and Lozzio, 1975*) lysates and rat brain cytosol (*Figure 3E*). This Nb thus recognizes the native full-length protein. And in agreement with Nb E_142 also interacting with EPS15R (*Figure 3A*), there is quantitative precipitation of EPS15R from the K-562 lysate. Analysis of co-immunoprecipitating components shows the known EPS15/R interaction partners FCHO1, FCHO2 and AP-2 (α subunits) (*Figure 2B*) are co-recovered with the EPS15/R immobilized through the EH domain regions by Nb E_142 (*Figure 3E*). The extent of recovery of cytosolic FCHO1/2, which bind to the C-terminal DPF region of EPS15/R with the µ-homology domain (µHD) (*Henne et al., 2010*; *Reider et al., 2009*; *Umasankar et al., 2012*), and AP-2, which binds to the DPF region of EPS15/R utilizing the appendages of the α and β2 subunits (*Benmerah et al., 1996*; *Iannolo et al., 1997*), mirrors the known differential affinities (*Ma et al., 2016*). In the same assay, NPF-dependent EH domain binding partners (HRB, *Figure 3E*) do not co-precipitate with Eps15/R, again in line with function-blocking Nb activity.

## Nb E_142 interacts with EPS15 intracellularly

Unexpectedly, when transiently transfected into HeLa cells, the diffuse, rather featureless intracellular distribution of an mNeonGreen (mNG)-tagged (*Shaner et al., 2013*) Nb E_142 fusion is not substantially different from the control mNG protein alone, and is clearly distinct from the punctate pattern of endogenous EPS15 staining in these cells detected with an anti-peptide antibody recognizing the extreme C-terminus (*Hinrichsen et al., 2003*) (*Figure 4A–E*). Immunoblots of whole cell extracts from transfected HeLa cells confirm synthesis and stability of the intact mNG-Nb fusion proteins as well as various clathrin coat components, including EPS15 (*Figure 4F*). In fewer than half the transfected cells, small concentrations of mNG-Nb E_142 that overlap some EPS15 puncta are seen (*Figure 4B'*, arrowheads). Essentially similar results are obtained irrespective of whether Nb E_142 is tagged at the N or C terminus with mNG, or with enhanced green fluorescent protein (GFP). In addition, in repeated experiments, transfection of either GFP-tagged Nb E_3 or Nb E_180 produces an

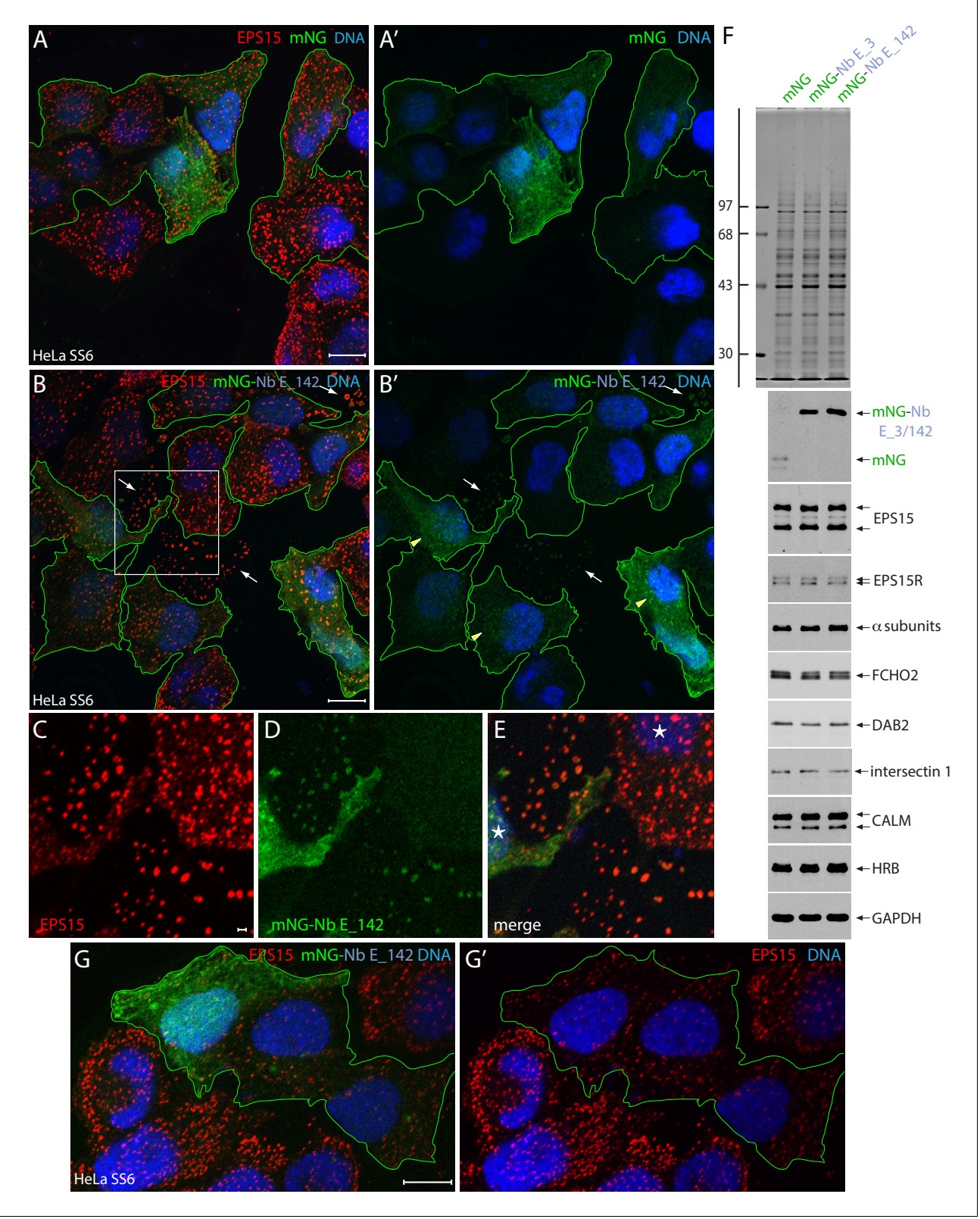

**Figure 4.** Intracellular expression of mNG-Nb E_142 in HeLa cells. (A–E) Selected but representative basal confocal optical sections of HeLa SS6 cells, fixed after transient expression of either control mNG (A and A'; green) or mNG-Nb E_142 (B–G), and stained with anti-EPS15 antibodies (red) and Hoechst DNA dye (blue). The mNG-expressing transfected cells are outlined (green) and glass-attached ruptured cell cortices (arrows) and Nb E_142 colocalizing weakly with endogenous EPS15 (arrowheads) in intact cells are indicated. Color-separated enlargements (C–E) of the region boxed in B are
*Figure 4 continued on next page*

*Figure 4 continued*

shown with nuclei (asterisks) in intact cells identified. The presence of mNG-Nb E_142 within EPS15-positive, morphologically aberrant clathrin-coated structures is evident on the adherent membrane fragments from sheared-off non-viable cells (*Heuser, 1989*). Scale bar; 10 µm; 1.0 µm in C. (F) SDS-PAGE and immunoblot analysis of whole HeLa cell lysates after transfection with plasmids encoding mNG, mNG-Nb E_3 or mNG-Nb E_142 as indicated and either stained with Coomassie blue or probed with the indicated antibodies. Molecular mass standards (in kDa) are shown on the left. (G–G′) Selected but representative confocal basal optical section of HeLa SS6 cells expressing mNG-Nb E_142 and analyzed as in B. mNG-expressing transfected cells are outlined. Scale bar; 10 µm.

DOI: https://doi.org/10.7554/eLife.41768.006

The following figure supplement is available for figure 4:

**Figure supplement 1.** Efficient intracellular labeling of endogenous EPS15 in AP-2-positive clathrin-coated structures with a polyvalent tandem mNG-Nb E_142 × 3 fusion protein.

DOI: https://doi.org/10.7554/eLife.41768.007

analogous dispersed location in HeLa cells. While superficially these results suggest Nb E_142 may not function effectively as a so-called chromobody (*Rothbauer et al., 2006*), in some mNG-Nb E_142-expressing cells an alteration in the steady-state distribution of endogenous EPS15 is evident compared with the adjacent non-transfected cells (*Figure 4G*). The intensity of the EPS15 surface puncta is diminished relative to adjacent non-transfected cells, indicative of less EPS15 deposition at these sites, and some limited interference with EPS15 function. Additionally, in adherent cell membrane cortices from inadvertently sheared-off cells (identified by lack of DNA staining), a dim mNG-Nb E_142 signal is seen to colocalize closely with endogenous EPS15 (*Figure 4B*, arrows, and *Figure 4C-E*). As Nb E_142 is able to robustly immunoprecipitate EPS15(R) from cytosol, and because the Nbs act as pseudoligands, these transfection data suggest that in viable cells EPS15/R deposited at clathrin coat assembly sites at the plasma membrane have EH domains locally occupied in a way that makes Nb E_142 binding unfavorable.

The concept of avidity-dependent, regionalized NPF occupancy of the EPS15/R EH domains gathered at incipient clathrin-coated structures is supported experimentally by the intracellular distribution and positioning of a transiently transfected triple Nb E_142 array. In HeLa cells, compared with the single Nb E_142 fusion, a larger mNG-Nb E_142 × 3 protein construct (*Figure 4—figure supplement 1*) now labels punctate AP-2-positive structures at very low expression levels. Given that both the ectopic mNG-Nb E_142 and the mNG-Nb E_142 × 3 proteins each contain only a single fluorescent mNG domain, the increased efficiency of EPS15/R labeling by the polyvalent reporter (*Figure 4—figure supplement 1*) must signify better apparent affinity for the massed EH domains at the sites of clathrin-coat assembly. This indicates that through avidity-based phenomena, artificially but flexibly grouped NPF-harboring Nbs can compete in vivo with the deployment multiple NPF-motifs by endogenous EH domain partner proteins.

Markedly different results are obtained when a single Nb E_142 is appended in-frame to a tandem Fab1p, YOTB, Vac1p, and EEA1 (FYVE$_2$) domain (*Figure 5A*); this leads to strong sequestration of endogenous EPS15 within cells. In HeLa SS6 cells, transient overexpression of the control GFP-FYVE$_2$ alone, which binds to the endosomal compartment phospholipid marker phosphatidylinositol 3-phosphate (PtdIns3P) (*Chen and De Camilli, 2005*), results in an enlarged labeled endosome compartment (*Figure 5B*), but the subcellular distribution of EPS15 remains undisturbed at clathrin-coated structures on the cell surface. A C-terminal Nb E_142 extension to the GFP-FYVE$_2$ chimera produces a dramatic alteration in steady-state EPS15 positioning; the protein now colocalizes extensively with the clustered GFP-FYVE$_2$-Nb E_142 fusion protein on large swollen endosomes and near completely disarranges assembly of EPS15 at typical scattered puncta on the plasma membrane (*Figure 5C*). To explain the markedly altered subcellular distribution of EPS15, note that a fundamental tenet of coat-assisted vesicle formation is the cycling of coat protomers between a membrane-attached, assembled pool and a cytosolic, reserve pool. If EPS15 is one of the early arriving pioneers at a nascent bud, it is deposited from a supply in the dispersed soluble pool. The population of EPS15 in the cytosolic fraction (*Figure 3*) (*van Delft et al., 1997a*) confirms that this coat protein too cycles between a membrane-assembled and a cytosolic pool. Additionally, like the tandem mNG-Nb E_142 × 3 fusion, a concentrating/multivalency effect of densely tethered Nb E_142 upon a two-dimensional membrane surface markedly increases the capacity of the Nb to capture endogenous EPS15 at that site. The endosome-bound Nb E_142 can thus lure EPS15 from the latter soluble

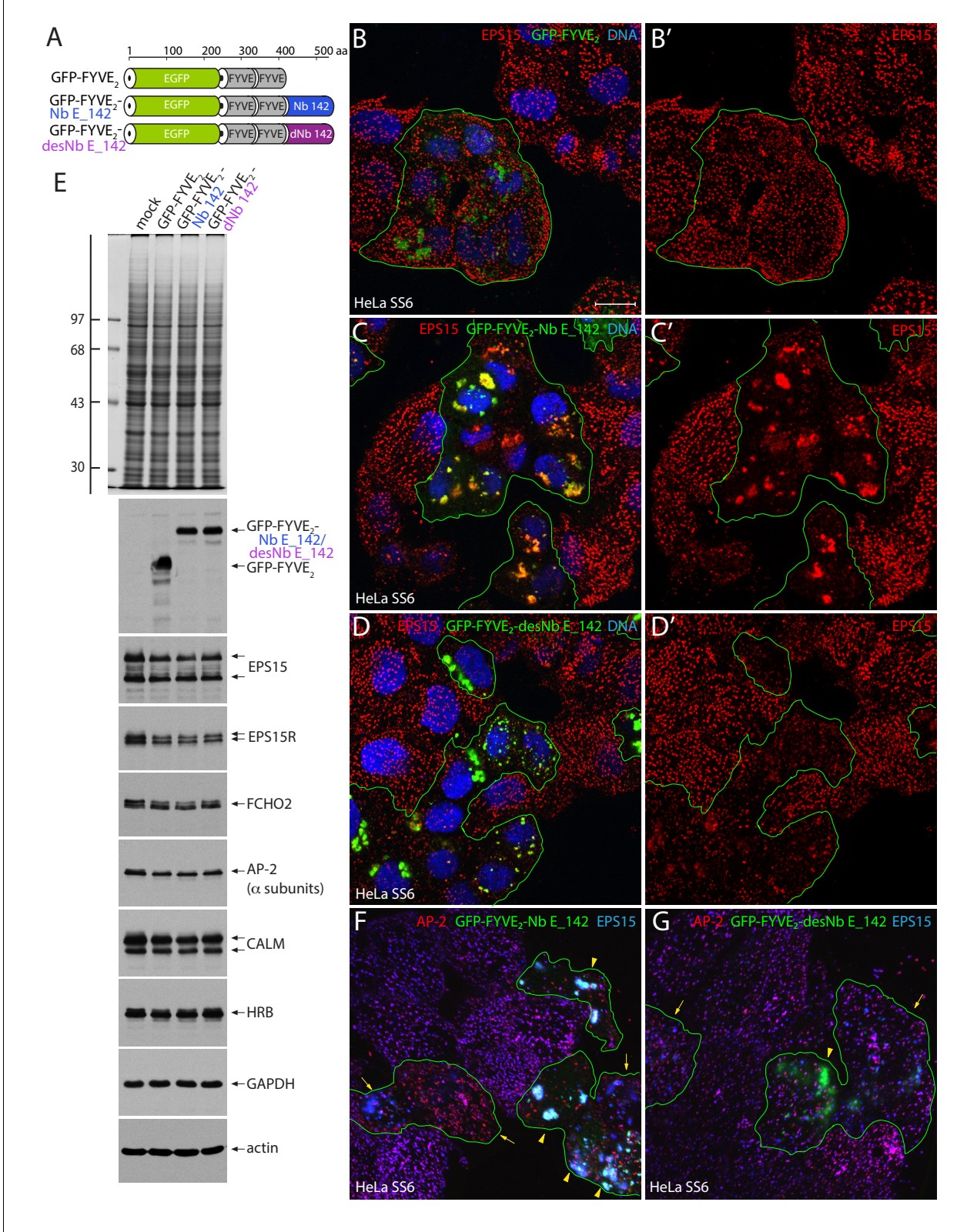

**Figure 5.** Massive endosome displacement of endogenous EPS15 with GFP-FYVE$_2$-Nb E_142. (**A**) Schematic depiction of the domain arrangement of the GFP-FYVE$_2$-based fusion proteins analyzed. (**B–D'**) Selected but representative confocal basal optical sections of HeLa SS6 cells transfected with either negative control GFP-FYVE$_2$ (**B, B'**), GFP-FYVE$_2$-Nb E_142 (**C, C'**) or destabilized GFP-FYVE$_2$-desNb E_142 (**D, D'**). Fixed cells were stained with anti-EPS15 antibodies (red) and DNA dye (blue). Scale bar; 10 μm. (**E**) SDS-PAGE and immunoblot analysis of whole HeLa SS6 cell lysates either mock

*Figure 5 continued on next page*

Figure 5 continued

transfected or transiently transfected with GFP-FYVE$_2$-based plasmids as indicated. Gels were either stained with Coomassie blue or transferred to nitrocellulose and probed with the indicated antibodies. Molecular mass standards (in kDa) are shown on the left. (F–G) Basal confocal section of transiently transfected HeLa SS6 cells expressing either GFP-FYVE$_2$-Nb E_142 (F) or GFP-FYVE$_2$-desNb E_142 (G) and stained for AP-2 α subunit (red) or EPS15 (blue). Moderate (arrows) and high-level expression (arrowheads) and EPS15 with AP-2 sequestration are indicated.
DOI: https://doi.org/10.7554/eLife.41768.008

The following figure supplement is available for figure 5:

**Figure supplement 1.** Endocytic transferrin uptake in GFP-FYVE$_2$-Nb E_142 expressing HeLa SS6 cells .
DOI: https://doi.org/10.7554/eLife.41768.009

pool, titrating out the available free EPS15 such that, over time, little is available for deposition at new clathrin bud sites. Thus organelle-attached Nb E_142 disturbs the steady-state balance of endogenous EPS15(R) positioning within the cell, and therefore does function as an intrabody.

Compared with the GFP-FYVE$_2$-Nb E_142 chimera, a destabilized mutant (designated 3maj; *Tang et al., 2016*) with three framework side chain substitutions (S73R, C98Y and S117F; GFP-FYVE$_2$-desNb E_142; see *Figure 1C*) also relocalizes endogenous EPS15 to an expanded, PtdIns3P-positive endosomal compartment in HeLa SS6 cells (*Figure 5D*). The rationale for the conditionally-destabilized mutant is that in the absence of antigen binding, the structurally perturbed Nb V$_H$H is labile and subject to protein quality control and accelerated proteasomal degradation (*Tang et al., 2016*). However, the effect of the conditionally destabilized Nb on EPS15 redistribution is not as penetrant as the native Nb. Assembly of endogenous EPS15 at plasma membrane clathrin-coated structures is still evident. Despite the dramatic change in intracellular localization of EPS15, immuno-blots show that ectopic expression of neither Nb E_142 nor desNb E_142 alters the overall abundance of EPS15, EPS15R or other core clathrin coat components (*Figure 5E*). Further, at low-to-moderate relative expression levels, AP-2 remains at dispersed puncta at the plasma membrane despite the massive relocation of EPS15(R) to GFP-FYVE$_2$-Nb E_142-positive intracellular endosomal structures (*Figure 5F*, arrows). In cells expressing high levels of GFP-FYVE$_2$-Nb E_142 with prominent enlarged PtdIns(3)P-positive endosomes, AP-2 becomes mislocalized with the EPS15 on these internal organelles and AP-2 puncta at the cell surface become less abundant (*Figure 5F*, arrowheads) but not in the GFP-FYVE$_2$-desNb E_142 producing HeLa cells (*Figure 5G*). Yet, in 5 min uptake assays of fluorescently-tagged transferrin, GFP-FYVE$_2$-Nb E_142-producing cells traffic the endocytic tracer into the lumen of swollen endosomes in cells irrespective of whether there is strong AP-2 clustering with the Nb bound EPS15 or not (*Figure 5—figure supplement 1*, arrowheads). Because of the dilated lumen of expanded endosomes in the presence of the overexpressed FYVE$_2$ module, the internalized transferrin signal is generally neither as compact nor intense as in the adjacent untransfected cells (*Figure 5—figure supplement 1G–J*); thus the results are confounded by overexpression-induced endosome morphology alterations. Nevertheless, despite the FYVE$_2$-provoked variable endosomal dimensions (a right-shifted volume distribution of tracer-labeled endosomes), quantifying integrated fluorescence intensity of the internalized transferrin tracer shows that prominent sequestration of EPS15(R) on endosomes by ectopic Nb E_142 clearly does not terminate clathrin-mediated endocytosis of transferrin in HeLa SS6 cells (*Figure 5—figure supplement 1J*).

## Mitochondria-directed Nb E_142 effects on CCS structure and function

Because transient overexpression of the PtdIns3P-binding tandem FYVE domain alters endosome morphology considerably and may perturb endosomal structure and dynamics in functional studies (*Gucwa and Brown, 2014*; *Hu et al., 2002*), a mitochondria-tethered alternative Nb E_142 fusion was evaluated. The tailored chimera is composed of a 31-amino acid N-terminal mitochondrial outer membrane targeting element from *M. musculus* Akap1 (*Ma and Taylor, 2008*) fused in-frame to Nb E_142 and then followed by infra-red fluorescent protein (iRFP; *Filonov et al., 2011*). Very similar near-quantitative intracellular redistribution of endogenous EPS15 in HeLa SS6 cells expressing this fusion protein, but not a control (with central FKBP12-rapamycin binding (FRB) domain instead of the Nb E_142 module), is evident (*Figure 6A and B*). EPS15 puncta disappear from the basal surface, and in medial confocal optical sections, cells expressing the Akap1-Nb E_142-iRFP do not display assembled EPS15 puncta at the cell margins, as seen in the adjacent untransfected or Akap1-FRB-iRFP transfected cells (*Figure 6B'*, arrows). But unlike GFP-FYVE$_2$-Nb E_142 expression, even in

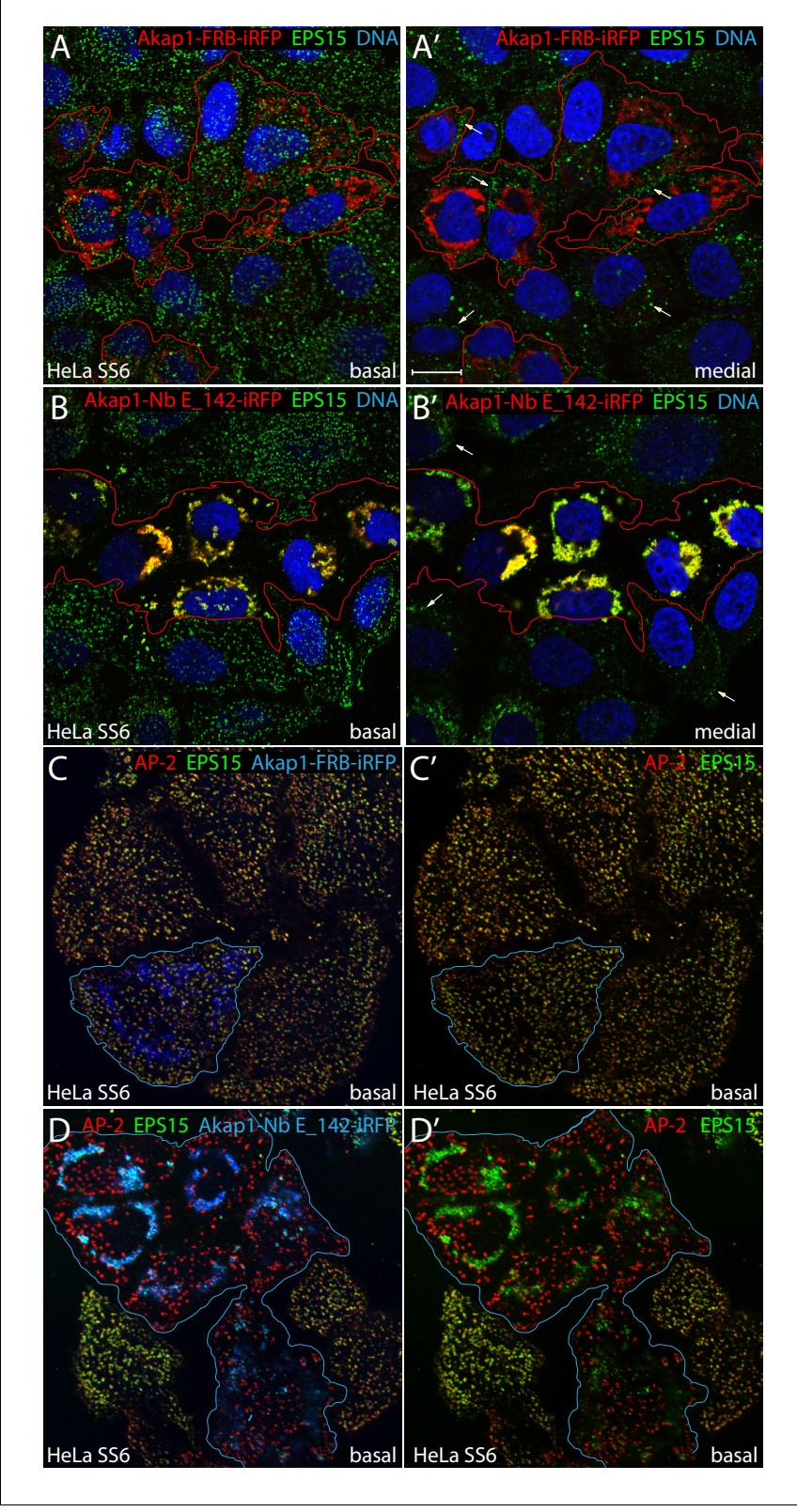

**Figure 6.** Mitochondrial capture of EPS15 with Akap1-Nb E_142-iRFP. (**A–B'**) Single confocal sections of basal (**A and B**) or medial (**A' and B'**) optical planes of fixed HeLa SS6 cells previously transfected with either control Akap1-FRB-iRFP (**A–A'**) or Akap1-Nb E142-iRFP (**B–B'**) (pseudocolored red) and stained with an antibody directed against EPS15 (green) and Hoechst DNA dye (blue). The iRFP-expressing transfected cells are outlined (red) and, in medial

*Figure 6 continued on next page*

*Figure 6 continued*

views, EPS15 assembled in clathrin-coated structures at the plasma membrane is indicated (arrows). (**C-D'**) Representative single basal confocal sections of fixed HeLa SS6 cells previously transfected with either control Akap1-FRB-iRFP (**C–C'**) or Akap1-Nb E142-iRFP (**D–D'**) (pseudocolored blue) and stained with anti-AP-2 α-subunit (red) and anti-EPS15 (green) antibodies. Relative colocalization of AP-2 with EPS15 in absence of Akap1-FRB/Nb E_142-iRFP signal (**C'** and **D'**) is shown. Scale bar; 10 μm.

DOI: https://doi.org/10.7554/eLife.41768.010

The following figure supplement is available for figure 6:

**Figure supplement 1.** Transferrin internalization in Akap1-Nb E_142-expressing HeLa cells.

DOI: https://doi.org/10.7554/eLife.41768.011

high level expression cells, the mitochondria relocalized EPS15 does not elicit prominent AP-2 reorganization, possibly reflecting the absence of a suitable source of phosphatidylinositol 4, 5-bisphosphate (PtdIns(4,5)P$_2$) support on the mitochondrial outer membrane and/or the lack of suitable cargo with appropriate sorting signals to bind to the AP-2 adaptor core (*Jackson et al., 2010*; *Traub and Bonifacino, 2013*) (*Figure 6C–D'*).

Functional studies show that both Akap1-FRB-iRFP and Akap1-Nb E_142-iRFP-producing HeLa cells internalize transferrin despite the conspicuous difference in EPS15 positioning within the Nb E_142-expressing cells (*Figure 6—figure supplement 1A–B'*), as is seen with the GFP-FYVE$_2$-Nb E_142-producing cells (*Figure 5—figure supplement 1*). A small subpopulation of Akap1-Nb E_142-iRFP-expressing cells display strongly suppressed, but not halted, transferrin internalization. Since a similar fraction (~15%) of the Akap1-FRB-iRFP fusion produces a similar phenotype, the diminished transferrin uptake is not likely due to mitochondrial dysfunction and ATP restriction but more probably due to the known cell-cycle dependence of transferrin endocytosis (*Sager et al., 1984*) and/or heterogeneity in relative internalization dependent upon local context and positioning of the cultured cells within the adherent population (*Snijder et al., 2009*). Overall, in the presence of a normal complement of endocytic clathrin coat components, artificial spatial displacement of EPS15/R onto mitochondria does not severely disrupt short-term clathrin-mediated endocytosis in parental HeLa SS6 cells.

## Mitochondria sequestration of EPS15 in gene-edited HeLa cells

As Eps15/R and Fcho1/2 function along the same pathway in a redundant manner (*Ma et al., 2016*; *Wang et al., 2016b*), the effect of forced expression of the mitochondria-targeted Nb E_142 was explored in novel HeLa cells rendered FCHO1/2- and EPS15R-null by successive TALEN-mediated gene editing (*Figure 7A*). Of the three new triple-null lines obtained that exhibit indels within the targeted exon 5 region of the EPS15R-encoding *EPS15L1* gene on chromosome 19 (*Figure 7B*), and have no detectable EPS15R protein on immunoblots (*Figure 7C*), one, clone #40, was selected for further analysis. The relative cellular abundance of EPS15, AP-2 and several other clathrin-mediated endocytosis components is not changed in the HeLa clone #40 cells (*Figure 7C*). Like the HeLa clone #64/1.E cells from which they are derived, clone #40 cells display morphologically abnormal clathrin-coated structures at steady state (*Figure 7D–F*); compared with the parental HeLa SS6 cells, fewer, larger and more clustered AP-2- and EPS15-positive clathrin assemblies are present on the ventral membrane in clone #40 cells. This abnormal distribution and morphology of clathrin-coated structures in the genome-edited cells implies FCHO1/2 and EPS15R play some significant role in bud site selection and maintenance.

Side-by-side comparison of transferrin uptake and the steady-state abundance of transferrin receptors at the cell surface reveals that both the HeLa #64/1.E (*Umasankar et al., 2014*) and the clone #40 cells display defective clathrin-mediated endocytosis compared with parental HeLa SS6 cells (*Figure 7—figure supplement 1*). Although fluorescent transferrin is internalized into peripheral early endosomes (lysosome-associated membrane glycoprotein 1 (LAMP1)-negative) in a 5 min pulse of the genome-edited cells at 37°C (accounting for their long-term viability), the extent is less than in the HeLa SS6 cells. Instead, the majority of the transferrin is bound to transferrin receptors diffusely situated over the plasma membrane (*Figure 7—figure supplement 1B'* and *C'*). This pool of surface transferrin receptors is not similarly evident in the parental cells. Comparative accumulation of surface transferrin receptors in the genome-edited lines is confirmed by fluorescent

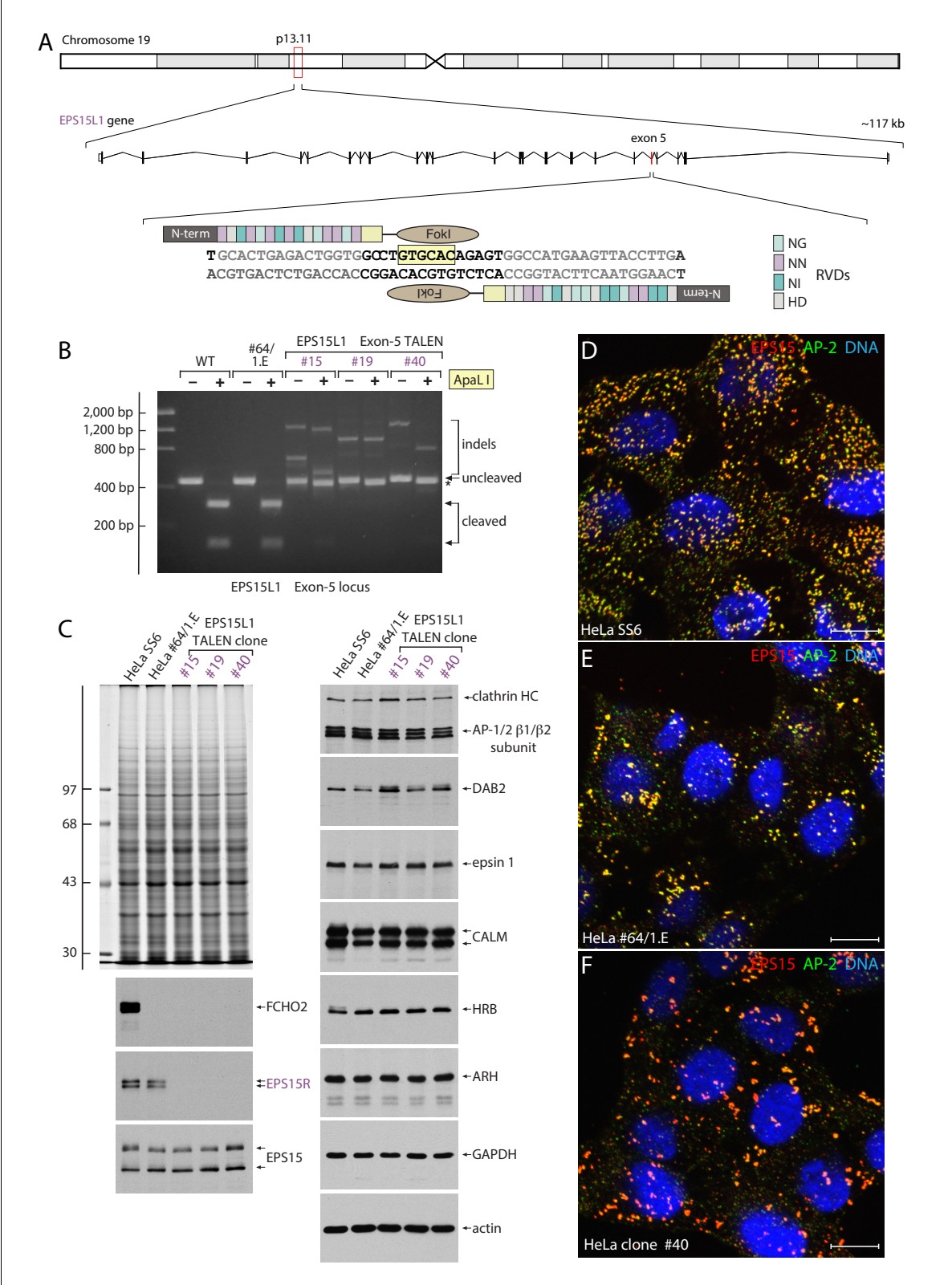

**Figure 7.** TALEN-mediated generation of FCHO1, FCHO2 and EPS15R triple-null HeLa cells. (**A**) Schematic depiction of the location and organization of the *EPS15L1* gene and TALEN design strategy. The assembled TALEN component repeat variable diresidues (RVDs) are shown, color-coded on the right. The diagnostic ApaLI restriction site (boxed in pale yellow) is indicated. (**B**) Agarose gel electrophoresis of *EPS15L1* exon 5 PCR products with and without ApaL1 digestion as indicated. Compared with the parental HeLa SS6 and clone #64/1.E cell PCR products, the abnormally large cleavage-

*Figure 7 continued on next page*

Figure 7 continued

resistant indels in the three TALEN clones is indicated. (C) SDS-PAGE and immunoblot analysis of TALEN clones. Whole cell lysates subjected to SDS-PAGE were either stained with Coomassie blue or immunoblotted with the indicated antibodies. Molecular mass standards (in kDa) are shown on the left. (D–F) Representative confocal optical section of the steady-state distribution and morphology of clathrin-coated structures at the surface of parental HeLa SS6 (D), HeLa clone #64/1.E (E) and HeLa clone #40 (F) cells stained with antibodies directed against EPS15 (red), the AP-2 adaptor α subunit (green), and Hoechst DNA dye (blue) as indicated. Scale bar; 10 μm.

DOI: https://doi.org/10.7554/eLife.41768.012

The following figure supplement is available for figure 7:

**Figure supplement 1.** Defective endocytic transferrin internalization in genome-edited HeLa cells.

DOI: https://doi.org/10.7554/eLife.41768.013

transferrin labeling experiments conducted on ice for 60 min (unpublished); the data thus indicate slowed egress of the transferrin receptor from the surface of the genome-edited cells for reentry into the endosomal compartment, leading to stalling and accumulation of this (and other) endocytic cargo at the plasma membrane.

Transient expression of negative control Akap1-FRB-iRFP in the HeLa clone #40 cells has no obvious effect on the subcellular patterning of EPS15 (*Figure 8A*), as in parental the HeLa SS6 cells. The high degree of colocalization of EPS15 with AP-2 at surface puncta is not different from adjacent untransfected cells. By contrast, in Akap1-Nb E_142-iRFP producing clone #40 cells, ectopic trapping of EPS15 on the mitochondria-massed Nb E_142 is clear, with a concomitant loss of EPS15 from the plasma membrane. In these transfected clone #40 cells, AP-2 clusters at the surface are diminished in size and number in a manner that correlates with the efficiency of EPS15 segregation at mitochondria (*Figure 8B-F*). The majority of AP-2-positive puncta that do remain are populated with detectable endogenous EPS15 protein (*Figure 8C–F*). Thus, in this genetic background stable membrane-associated AP-2 assemblies depend heavily on EPS15 for their generation.

Based on the apparent affinity and computed cellular abundance in HeLa cells (*Hein et al., 2015*; *Itzhak et al., 2016*), HRB, HRB-like protein (HRBL) and DAB2 are most likely the predominant interaction partners for the EH domains at assembling clathrin-coated structures (*Figure 2*). Comparison of the steady-state distribution of EPS15 and HRB in clone #40 cells shows that whereas both proteins overlap at the aberrant enlarged clathrin-coated structures that are a distinctive feature of untransfected cells, in Akap1-Nb E_142-iRFP-expressing cells, neither protein is efficiently deposited on the plasma membrane (*Figure 8G–L*). In these cells, EPS15 is lured onto the cytosolic face of mitochondria but HRB is not, and EPS15 is associated with the few HRB puncta that are evident at the plasma membrane (*Figure 8I–L*). Thus, in the presence of Akap1-Nb E_142-iRFP, HRB is essentially displaced into the soluble cytosolic compartment. Overall, these findings recapitulate the effect of combined EPS15 and EPS15R RNAi in the parental HeLa clone #64/1.E cells, where there is a shift from large irregular clathrin-coated assemblies to small, abortive diffraction-limited structures (*Ma et al., 2016*).

The defects in AP-2-containing clathrin-coated structures in Akap1-Nb E_142-iRFP-transfected clone #40 cells translate into clathrin-mediated endocytosis dysfunction. A modified uptake assay format (*Reis et al., 2015*), with a 2 min pulse of transferrin at 37°C followed by surface stripping on ice and rewarming to 37°C for a further 2 min (*Figure 9A*), was used since the clone #40 cells display faulty clathrin-mediated internalization (*Figure 7—figure supplement 1*). As this procedure removes the surface bound transferrin pool, alterations in rapid internalization are easier to discern. Loss of EPS15 from surface puncta in clone #40 cells by massing upon mitochondria with Akap1-Nb E_142-iRFP has a strong inhibitory effect on transferrin uptake (*Figure 9*). In both basal and medial confocal optical sections and maximal intensity projections, the presence of small intracellular endosomes positive for the fluorescent endocytic tracer is strongly decreased compared with the neighboring untransfected cells, and also with control Akap1-FRB-iRFP producing clone #40 HeLa cells (*Figure 9*). In these experiments, quantitation shows that the number of >0.2 μm$^2$ transferrin-positive intracellular structures within the three-dimensional volume of untransfected cells surrounding the Akap1-Nb E_142-iRFP-expressing cells is roughly double that of the EPS15-sequestered cell population (*Figure 9—figure supplement 1*). Since the larger transferrin-containing vesicular structures represent progressive concentrating endosomal fusion reactions (*Rink et al., 2005*), and because the Akap1-Nb E_142-iRFP-transfected HeLa clone #40 cells display substantially elevated levels of transferrin

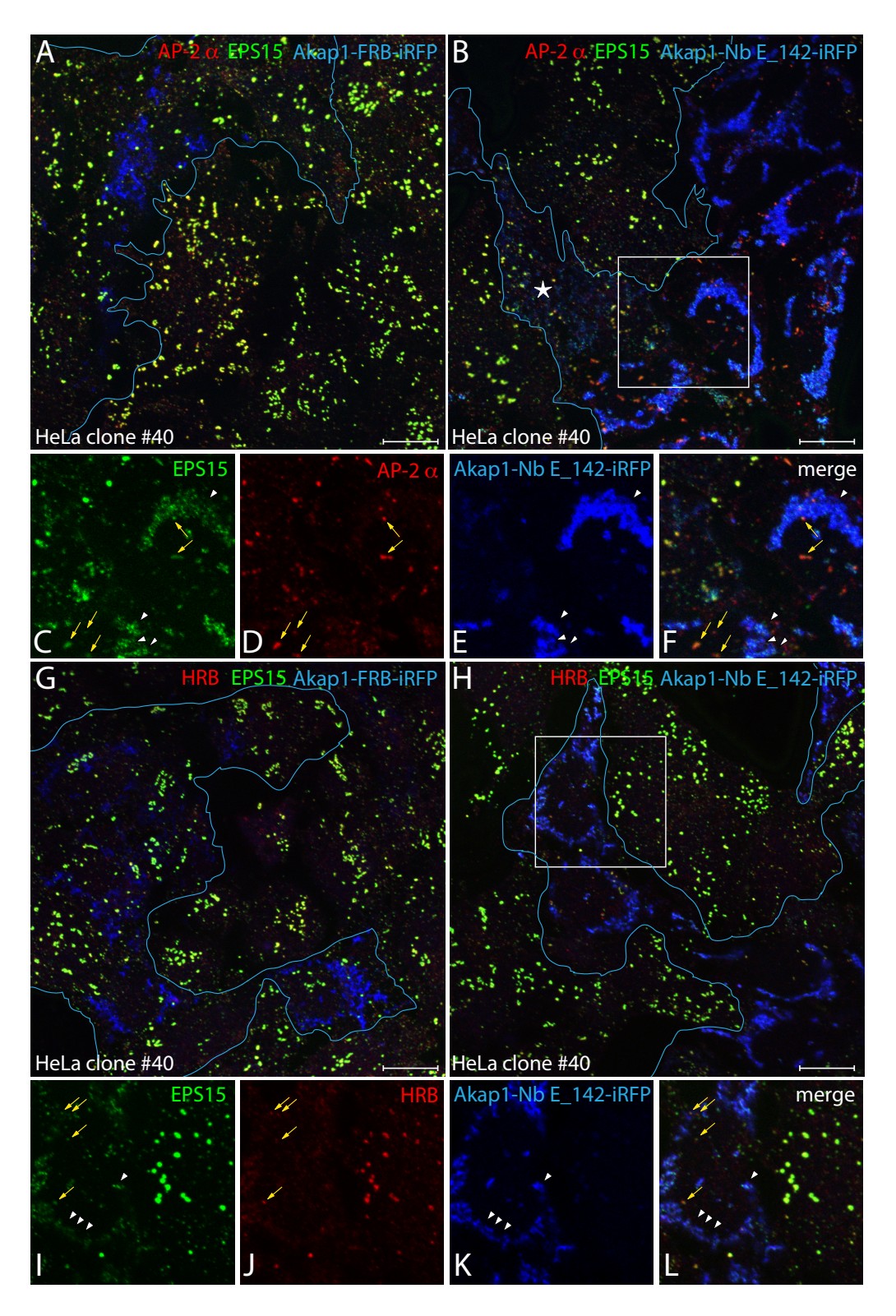

**Figure 8.** Clathrin-coated structure abnormalities in HeLa clone #40 cells lacking FCHO1/2 and EPS15/R. (**A–L**) Representative confocal basal optical sections of fixed HeLa clone #40 cells expressing either control Akap1-FRB-iRFP (**A and G**) or Akap1-Nb E_142-iRFP (**B-F, H-L**; transfected cells outlined in blue) and stained with antibodies to EPS15 (green) and the AP-2 α subunit (red) or HRB (red), as indicated. Color-separated enlargements of the
*Figure 8 continued on next page*

*Figure 8 continued*

regions boxed in **B** and **H** are shown below, with colocalization of displaced endogenous EPS15 with Nb E_142 (arrowheads) in transfected cells and AP-2 with residual EPS15 or HRB at the plasma membrane (arrows) indicated. Scale bar; 10 μm.

DOI: https://doi.org/10.7554/eLife.41768.014

receptors at the cell surface (*Figure 7—figure supplement 1*), this inhibited internalization with mitochondria-appropriated EPS15 implies that FCHO1/2 and EPS15/R facilitate AP-2-dependent clathrin-mediated endocytosis.

## Rescue of AP-2 puncta in clone #40 cells with grossly misplaced EPS15 by the FCHO1 unstructured interdomain linker

Ectopic expression of Akap1-Nb E_142-iRFP is unlikely to grossly perturb PtdIns(4,5)P$_2$ levels on the basis of the proper plasma membrane deposition of endogenous AP-2 in transfected HeLa SS6 cells (*Figure 6*). Indeed, the failure of AP-2 to assemble into normally dispersed puncta at the surface of clone #40 cells is not due to bulk effects on phosphoinositide metabolism in Akap1-Nb E_142-iRFP-transfected cells. Using the phospholipase Cδ1 pleckstrin-homology (PH) domain as an intracellular biosensor for PtdIns(4,5)P$_2$, both co-transfected Akap1-FRB-iRFP or Akap1-Nb E_142-iRFP-expressing cells show an abundance of this plasma membrane localized lipid even with the marked difference in EPS15 localization (*Figure 10A and B*). These results reveal that despite a plasma membrane supply of PtdIns(4,5)P$_2$, elevated levels of the transferrin receptor at the cell surface (*Figure 7—figure supplement 1*), and a normal cellular concentration of AP-2 (*Figure 7C*), the absence of utilizable FCHO1/2 and EPS15/R pioneers in HeLa clone #40 cells is not compatible with widespread and stable deposition of AP-2 at the cell surface and assembly of clathrin-coated structures. In addition, the noted parallel decrease in AP-2 and HRB puncta in the presence of the Akap1-Nb E_142-iRFP trap again supports the notion that the EH domains of EPS15/R play a key role in docking HRB at nascent coated buds.

Fully concordant with the widely dispersed PtdIns(4,5)P$_2$ within the inner leaflet of the plasma membrane in both Akap1-FRB-iRFP- and Akap1-Nb E_142-iRFP-transfected HeLa clone #40 cells is the restoration of extensive and apparently random deposition of AP-2 on co-transfection with GFP-tagged FCHO1 (1–609) (*Figure 11*). This truncated FCHO1 protein contains the membrane-binding N-terminal EFC/F-BAR domain (*Umasankar et al., 2012*; *Henne et al., 2007*) and the adjoining intrinsically disordered linker that appears to operate as an allosteric activator (*Umasankar et al., 2014*; *Hollopeter et al., 2014*), driving a reconfiguration of AP-2 from the closed to the membrane-bound open state (*Jackson et al., 2010*). Ectopic GFP-FCHO1 (1–609) can reconstitute the regularly dispersed HeLa clathrin-coated structure pattern at very low levels of expression of the linker (*Figure 11A*). Measurement of AP-2-positive puncta in FCHO1 linker-expressing cells, compared with adjacent non-transfected HeLa clone #40 cells, documents the overall shift to smaller, more numerous 0.2–0.6 μm$^2$ structures (*Figure 11—figure supplement 1*). In parallel technical replicates, as expected EPS15 is sequestered prominently by the overexpressed Nb decoy on mitochondria, and little endogenous EPS15 is deposited along with the corrected AP-2 in the compact, scattered surface puncta (*Figure 11B*). Thus the normalizing effect of the FCHO1 (EFC domain and) linker is manifest largely in the absence of co-assembled EPS15. Also in the same experiment, by itself the expressed GFP-FCHO1 (1–609) produces a profusion of small clathrin-coated structures spread over the basal plasma membrane that contain both AP-2 and EPS15, while the transfected Akap1-Nb E_142-iRFP alone repeatedly diminishes the intensity and abundance of AP-2 structures at the ventral surface of clone #40 cells (*Figure 11C–D*). The FCHO1-corrected phenotype is starkly different from the untransfected clone #40 cells, and closely mirrors the results seen in clone #64/1.E cells (*Umasankar et al., 2014*; *Sager et al., 1984*), where we have shown this is not due to the phospholipid-binding N-terminal EFC domain (*Umasankar et al., 2014*). These striking results again verify that FCHO1/2 act epistatically to EPS15/R and demonstrate clearly that a major role of EPS15/R is normally to cooperate with full-length FCHO1/2 to bind to and activate AP-2 proteins at incipient clathrin-coated structures, as we previously suggested (*Ma et al., 2016*).

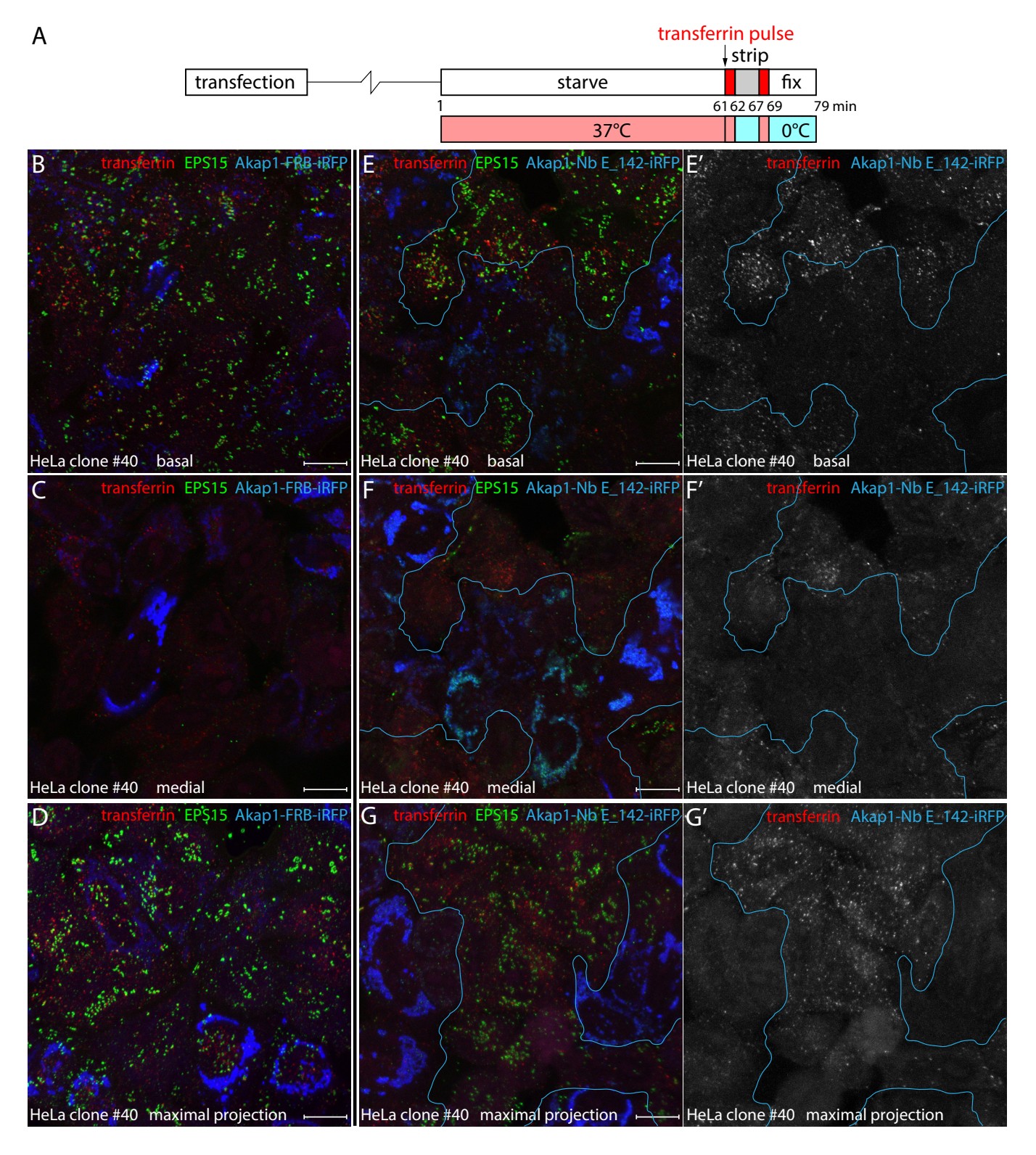

**Figure 9.** Defective clathrin-mediated endocytosis in Nb E_142-producing clone #40 cells. (A) Schematic representation of the transient transfection and transferrin internalization protocol. (B-G') Transiently transfected HeLa clone #40 cells expressing either negative control Akap1-FRB-iRFP (B-D) or Akap1-Nb E_142-iRFP (E–G') were incubated in starvation medium for 60 min before a 2 min pulse of 25 µg/ml Alexa Fluor 568-transferrin (red; white in E'-G') at 37˚C. The cells were then placed on ice and immediately washed three times with acid stripping buffer to remove surface bound transferrin.
*Figure 9 continued on next page*

*Figure 9 continued*

Then 37°C medium was readded and the cells incubated for a further 2 min at 37°C before fixation on ice. Permeabilized cells were stained with antibodies against EPS15 (green), and confocal optical sections from a basal (**B and E**) and medial (**C and F**) plane or a maximal intensity projection of alternative fields (**D and G**) are shown with the iRFP fusion protein pseudocolored (blue). Isolated grayscale images of the transferrin (red) channel in the Akap1-Nb E_142-iRFP-transfected preparations are shown on the right (**E'–G'**) to better visualize the defective clathrin-mediated endocytosis. Scale bar; 10 μm.

DOI: https://doi.org/10.7554/eLife.41768.015

The following figure supplement is available for figure 9:

**Figure supplement 1.** Quantitation of short-pulse transferrin uptake in Akap1-Nb E142-iRFP transfected HeLa clone #40 cells.

DOI: https://doi.org/10.7554/eLife.41768.016

## Nb-E3 ligase mediated EPS15 degradation

An alternative experimental strategy to proximity and local molarity changes through Nb-mediated intracellular sequestration is targeted elimination of Nb-recognized targets (*Moutel et al., 2016*; *Gross et al., 2016*; *Shin et al., 2015*). The GFE3 plasmid, which contains a GFP reporter and an antibody-like gephyrin recognizing FingR module for ubiquitin conjugation and targeted proteasomal degradation of gephyrin mediated by an in-frame fusion of the *Rattus norvegicus* X-linked inhibitor of apoptosis (XIAP) E3 ligase RING domain (residues 440–496) (*Gross et al., 2016*), was repurposed by substitution of Nb E_142 for the fibronectin-derived FingR section. The negative control, parental GFE3 transfection into HeLa SS6 cells, has negligible impact on the steady-state distribution of EPS15 at clathrin-coated puncta on the ventral cell surface (*Figure 12A*). By contrast, the G-Nb E_142-E3 plasmid is highly effective in promoting the degradation of EPS15 in transiently transfected cells positive for GFP (*Figure 12B*). The effectiveness of the XIAP E3 is superior to similar constructs containing the C-terminus of Hsc70-interacting protein (CHIP) or von Hippel-Lindau tumor suppressor E3 ligase domain (data not shown). Variability of different synthetic E3 ligase chimeras to promote degradation of selected target proteins has been seen by others (*Shin et al., 2015*). As with Akap1-Nb E_142-iRFP, effective elimination of EPS15/R by the tailored G-Nb E_142-E3 fusion protein still leaves AP-2 assembled at clathrin-coated sites on the plasma membrane of the HeLa SS6 cells (*Figure 12C–D*).

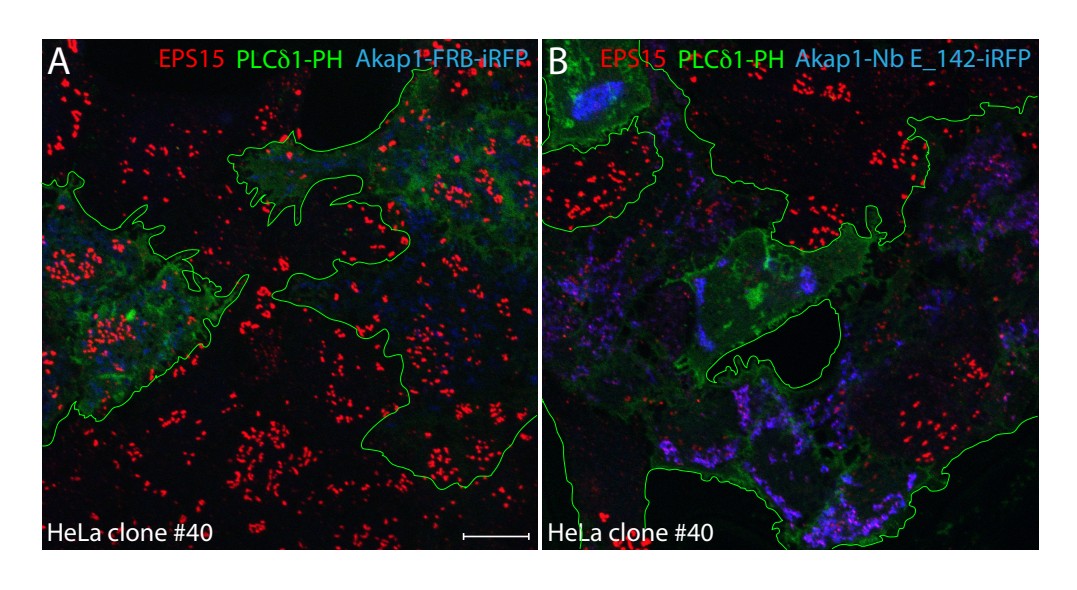

**Figure 10.** PtdIns(4,5)P$_2$ synthesis in clone #40 cells expressing mitochondria-targeted Nb E_142. (**A–B**) Single basal optical sections of HeLa clone #40 cells co-transfected with control Akap1-FRB-iRFP (**A**) or Akap1-Nb E_142-iRFP (**B**) (blue) together with PLCδ1 PH-GFP (green; transfected cells outlined). Fixed cells were probed with anti-EPS15 antibodies (red) as indicated. Scale bar; 10 μm.

DOI: https://doi.org/10.7554/eLife.41768.017

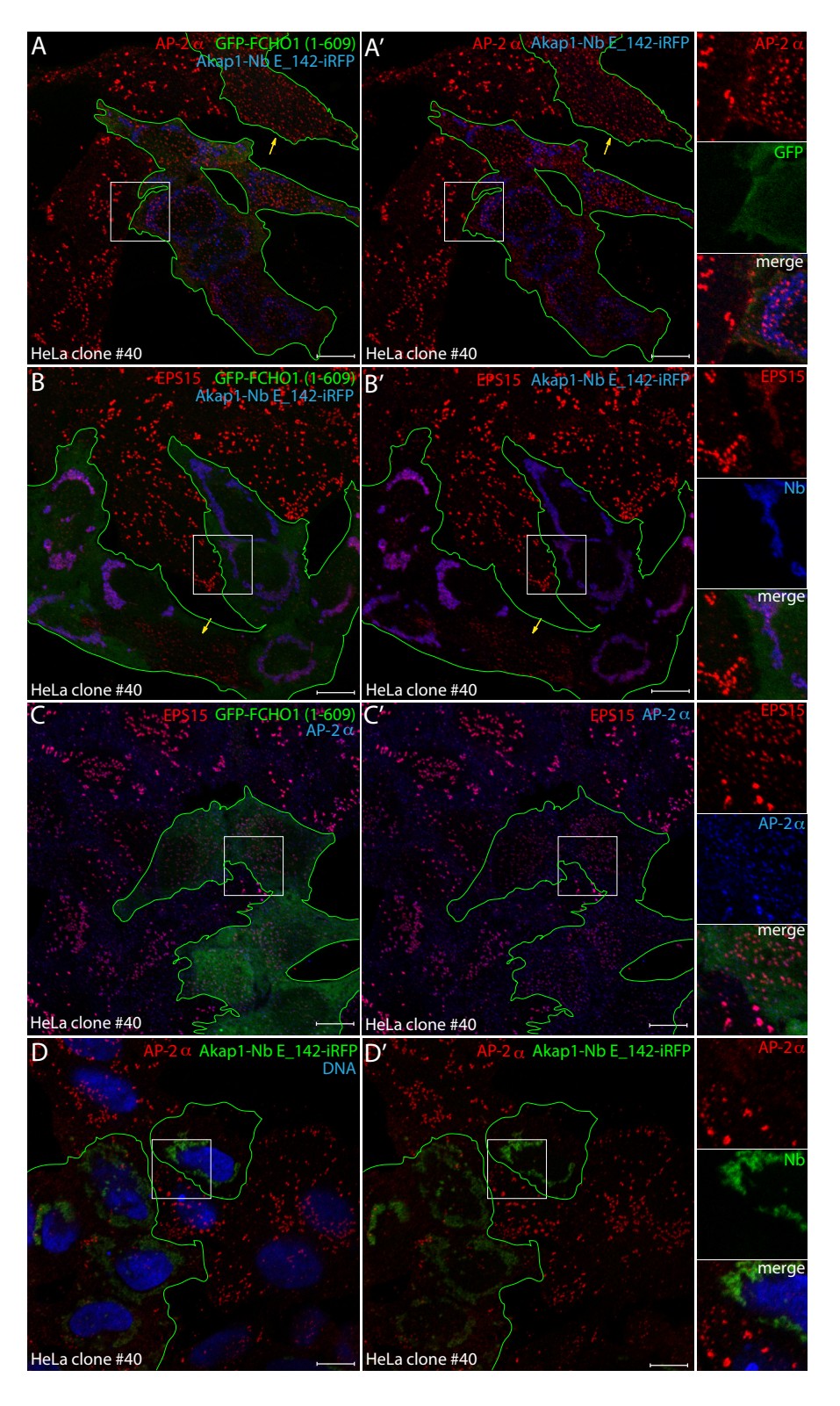

**Figure 11.** Reconstitution of clathrin-coated structure morphology in clone #40 cells with the FCHO1 disordered interdomain linker. (A–B') Representative confocal basal optical sections of HeLa clone #40 cells transiently transfected with both GFP-FCHO1 (1–609)- and Akap1-Nb E_142-iRFP-encoding plasmids before fixation. Permeabilized cells were stained with antibodies directed against either the AP-2 α subunit (red in A) or EPS15 (red in B). Because of diminished GFP expression in these co-transfected cells GFP was enhanced with the GFP-Booster Nb (green; transfected cells

*Figure 11 continued on next page*

*Figure 11 continued*

outlined), while the iRFP is pseudocolored (blue). Individual cells transfected with the GFP-FCHO1 (1–609) plasmid alone, lacking the mitochondrial iRFP signal, are indicated (yellow arrows), where the morphology of the clathrin-coated structures is rectified by the ectopic FCHO1 interdomain linker. Color-separated enlargements of the regions boxed in **A** and **B** are shown on the right. (**C–D′**) Basal optical sections of HeLa clone #40 cells transfected with either the GFP-FCHO1 (1–609) (**C**) or Akap1-Nb E_142-iRFP (**D**) plasmids alone before fixation and permeabilization. The GFP-FCHO1 (1–609)-expressing cells (green; transfected cells outlined) were stained with antibodies directed against the AP-2 α subunit (blue) and EPS15 (red), while the Akap1-Nb E_142-iRF-expressing cells were stained with anti-AP-2 α-subunit antibodies (red) and Hoechst DNA dye (blue) as indicated. Color-separated enlargements of the regions boxed in **C** and **D** are shown on the right. Scale bar; 10 μm.
DOI: https://doi.org/10.7554/eLife.41768.018

The following figure supplement is available for figure 11:

**Figure supplement 1.** Quantitation of clathrin-coated structure restructuring in GFP-FCHO1 (1–609) transfected HeLa clone #40 cells.
DOI: https://doi.org/10.7554/eLife.41768.019

## Discussion

In depth characterization of the anti-EPS15 Nb E_142 shows this $V_HH$ is a versatile reagent in that it functions both in in vitro biochemical assays and as an effective intrabody in the reducing environment of the cell cytosol. The molecular mode of action of function-blocking intrabodies varies. Some can disrupt a critical protein–protein, protein–nucleic acid or protein–lipid interaction, thereby preventing the normal biological activity of the target, or stabilize a specific, functionally significant conformational state (*Staus et al., 2016*; *Zimmermann et al., 2018*). Alternatively, and especially for proteins that operate as interaction hubs like Eps15/R (*Figure 2B*), where a single Nb cannot possibly incapacitate all of the interaction information encoded within the target protein, imposed regional compartmental sequestration is a companion strategy. Here, using new NPF motif-harboring llama Nbs, it is demonstrated that during assembly the collective EH domains of EPS15 and EPS15R deposited at clathrin-coated structures appear poorly accessible; this provides an important clue to the occupancy of EH domains during lattice assembly and sorting in light of the known modest $K_D$ for a single NPF•EH domain interaction (*Yamabhai et al., 1998*; *de Beer et al., 1998*). These results confirm and extend previous data showing that ectopic expression of the tandem EH domain module can inhibit clathrin-mediated endocytosis (*Carbone et al., 1997*), and that removal of EH2 + EH3 from EPS15 prevents proper targeting to and stationing at clathrin-coated structures (*Benmerah et al., 1999*). Inactivating point mutations to each of the three EH domains of the full-length *Drosophila* Eps15 orthologue abolish proper positioning at the synapse, while a C-terminally truncated ΔDPF mutant still localizes properly and is able to effectively rescue the defective synaptic vesicle endocytosis at *eps15*-null nerve terminals (*Koh et al., 2007*). And in the squid giant axon, microinjection of an NPF peptide interferes with clathrin-mediated endocytosis at an early stage, roughly halving the occurrence of clathrin-coated structures at the synapse (*Morgan et al., 2003*). Eps15b, an alternatively-spliced isoform of Eps15 in vertebrates, lacks the N-terminal EH1-3 tandem array entirely and does not localize to the plasma membrane but rather to endosomes (*Roxrud et al., 2008*).

Yet the EH domain module, alone, is not able to concentrate at clathrin-coated structures (*Benmerah et al., 2000*). In the budding yeast *S. cerevisiae*, where *Hs* EPS15 can substitute for the orthologous Ede1p, the minimal region of Ede1p necessary for proper targeting to clathrin-coated buds is EH3, a flanking proline-rich region and the coiled-coil domain (*Lu and Drubin, 2017*). This argues that EH domains play a critical role in placing Eps15 at the membrane bud site but highlights the cooperativity of the binding information hardwired into Eps15/R (*Figure 2B*) and their early-arriving binding partners. In metazoa, these include AP-2, Fcho1/2, Hrb, Dab2, epsin, and CALM (*Manna et al., 2015*). These endocytic proteins display a densely wired interactome, and it seems likely they are responsible for high frequency occupancy of the EH domains at clathrin assembly sites. Like EPS15/R, HRB/L (also designated AGFG1/2) and DAB2 are both detected in proximity labeling mass spectrometry experiments as early, and thus edge positioned, components (*Paek et al., 2017*). Because Nb E_142 can bind efficiently to EPS15/R in the soluble state, the poor labeling at clathrin-coated structures with mNG-Nb E_142 and other tagged versions indicates not only that the EH domain module is locally occupied at these assembly zones, but that any monomeric Nb E_142 bound to incoming EPS15/R is promptly displaced by the forming EH network that

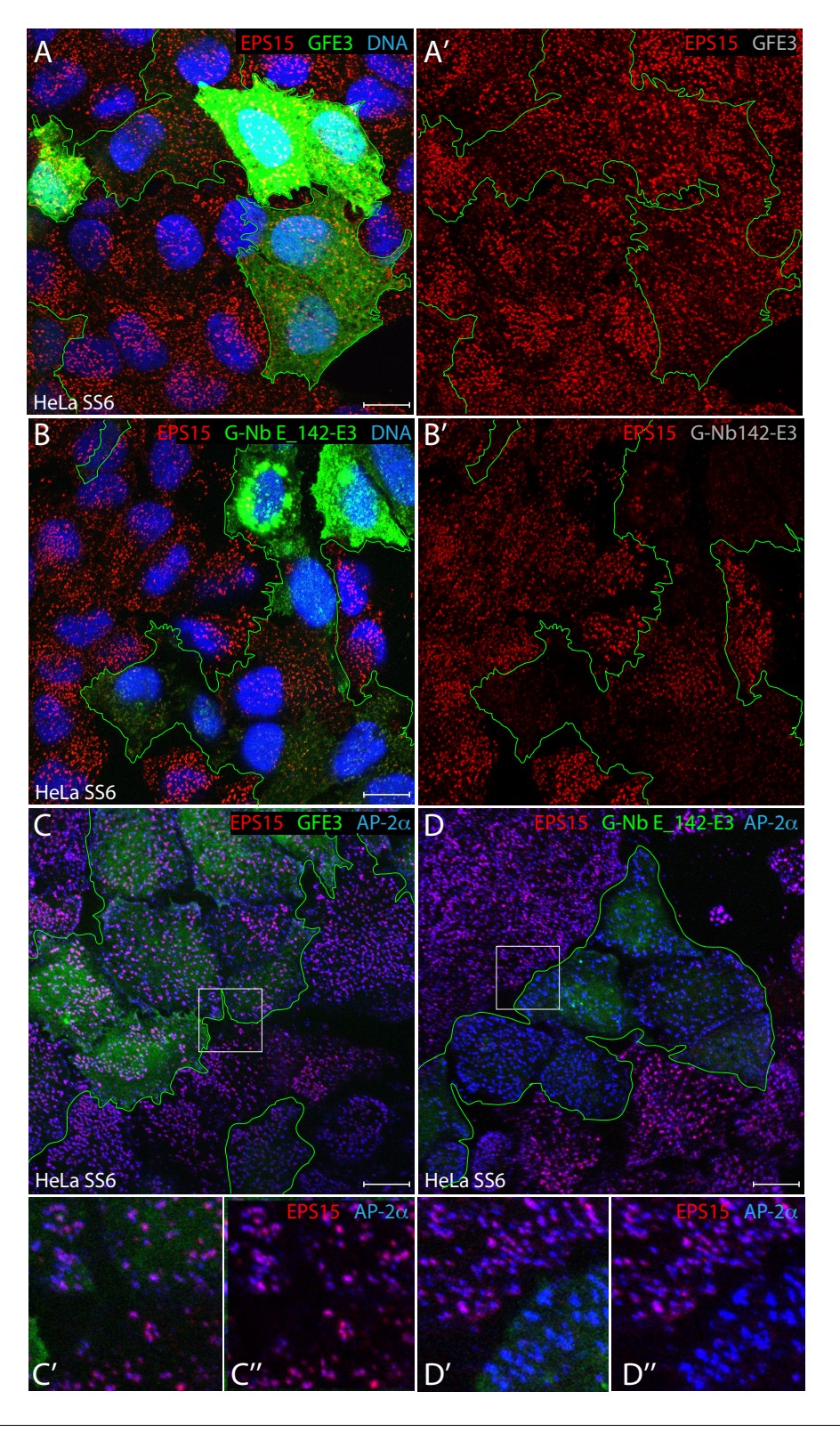

**Figure 12.** Ubiquitin-dependent degradation of EPS15 utilizing a repurposed E3 ligase. (A–B') Representative confocal basal optical sections of HeLa SS6 cells transfected with either GFE3 or G-Nb E_142-E3 plasmids before fixation and staining with anti-EPS15 antibodies (red), and Hoechst DNA dye (blue) as indicated. Cells expressing GFP-tagged E3 ligase fusions are outlined (green), with the separated EPS15 signal (A', B') shown. (C–D'') Fixed

*Figure 12 continued*

HeLa SS6 cells, transiently XIAP E3 ligase transfected as in **A** and **B** above, were stained with antibodies against EPS15 (red) and the AP-2 α subunit (blue) as indicated. Color-separated enlargements (**C′–D′′**) of the regions boxed in **C** and **D** are shown below. Scale bar; 10 μm.

DOI: https://doi.org/10.7554/eLife.41768.020

accompanies coat formation. Somewhat contradictory, Nb E_142 can compete with dilute binding partners in vitro (*Figure 2*); Nb E_142 therefore varies in the ability to bind stably to EH domains dependent on the surrounding microenvironment. A plausible explanation for this apparent discrepancy is an increase in effective membrane concentration of the endogenous EH binding partners at nascent clathrin bud sites by two-dimensional partitioning and avidity effects slowing unbinding and diffusion away (*Yogurtcu and Johnson, 2018*).

The results from the new triple-null clone #40 HeLa cells expressing a mitochondria-bound Nb E_142 illustrate that without FCHO1/2 and EPS15/R, coated assemblies do not form stably and clathrin-mediated endocytosis falters. Thus the steady-state perdurance of clathrin-coated structures depends on presence of the pioneer EPS15 in clone #40 cells, albeit with abnormal morphology. However, in parental HeLa cells, loss of either set alone does not similarly abolish uptake (*Ma et al., 2016*; *Umasankar et al., 2012*; *Teckchandani et al., 2012*) (*Figure 5—figure supplement 1*, *Figure 6—figure supplement 1*). To explain this compensation, there must be a shared partner protein (s) that is bound to both these pioneers, and can still operate in the absence of one. Again, this is emblematic of the extensive redundancy in the pioneer interaction web (*Figure 2B*). Potential candidates for a critical common partner are once more AP-2 and/or its direct binding partners DAB2, CALM, HRB or epsin. Both CALM and HRB operate as vital SNARE retrieving sorting adaptors (*Miller et al., 2011*; *Pryor et al., 2008*). That the Nb-directed mislocalization of EPS15 in the HeLa clone #40 cells leads to bulk loss of clathrin-coated structures and diminished clathrin-based uptake indicates that the EH domain proteins intersectin 1 and intersectin 2, which are both expressed in HeLa cells (*Figure 4*) (*Itzhak et al., 2016*), cannot sustain the process without FCHO1/2 and EPS15/R. These findings differ from siRNA-based experiments (*Teckchandani et al., 2012*) unless Nb E_142 also quantitatively depletes intersectins. Nb E_142 does not bind physically to the intersectin 1 EH domains, and although hierarchical clustering, based on ultrastructural localization within clathrin-coated structures, places EPS15/R and intersectin in distinct branches (*Sochacki et al., 2017*), and the BioPlex 2.0 database (with HEK 293 cells) (*Schweppe et al., 2018*) does not score EPS15/R as intersectin 1 or -2 binding partners, the noted discrepancy could be due to intersectins binding physically to EPS15/R (*Sengar et al., 1999*) and so being concomitantly unavailable when massed at an inappropriate intracellular organelle, or due to lower cellular abundance. Quantitative proteomics reveals that in HeLa cells intersectin 1 and -2 are present at ~20–30% the level of EPS15/R (*Hein et al., 2015*; *Itzhak et al., 2016*).

The experiments presented here reveal that the pioneers localized to the perimeter of clathrin-coated structures at steady state (*Sochacki et al., 2017*; *Ma et al., 2016*; *Tebar et al., 1996*; *Henne et al., 2010*; *Edeling et al., 2006*) are generally necessary for the inception of new clathrin-coated buds. Because HeLa clone #40 cells clearly have PtdIns(4,5)$P_2$ in the inner leaflet of the plasma membrane when Akap1-Nb E_142-iRFP is expressed, the diminished number and/or intensity of AP-2 puncta in these cells highlights the supportive role the FCHO1/2-EPS15/R partnership plays in stabilizing active, conformationally-rearranged AP-2 at the cell surface. The still normal abundance of cytosolic AP-2 and clathrin in these multiply genome-edited cells tends to argue strongly against AP-2 pairs simply engaging the PtdIns(4,5)$P_2$-rich plasma membrane and simultaneously interacting with a clathrin triskelion to initiate stable, productive clathrin coat assembly (*Cocucci et al., 2012*). Indeed, super-resolution microscopy of the AP-2 and clathrin distribution at the ventral surface in single HeLa cells reveals substantial amounts of membrane-deposited AP-2 without a coincident clathrin signal (*Leyton-Puig et al., 2017*). While it is possible that a priming FCHO1/2•EPS15/R nanocluster (*Ma et al., 2016*) operates downstream or alongside AP-2 colliding with the plasma membrane in a suitable orientation to engage PtdIns(4,5)$P_2$ through the binding sites on the large α- and β2-subunit trunks (*Jackson et al., 2010*; *Gaidarov et al., 1996*), the fact that the intrinsically disordered linker in Fcho1/2 appears to act as a regulator of AP-2 switching from the closed to the open state (*Umasankar et al., 2014*; *Hollopeter et al., 2014*) suggests that

these pioneers promote the stable association of AP-2 with the membrane. The strongest evidence for this model is the unmistakable reconstitution of apparently normal AP-2 puncta in the presence of a membrane-tethered FCHO1 linker domain in clone #40 cells (*Figure 11*). Under normal circumstances at the edge of a de novo forming coat, membrane-attached FCHO1/2 interacts with incoming EPS15/R through the µHD (*Ma et al., 2016*; *Shimada et al., 2016*), steering the AP-2-bound EPS15 to relay the heterotetramer to FCHO1/2 for local access to the activating linker. This concept is in line with EPS15/R being the highest affinity binding partner of the FCHO1/2 µHD (*Umasankar et al., 2012*; *Shimada et al., 2016*) and being the only proteins in the genome with the tandem $DPFX_{2-3}DPFX_{2-3}DPF$ SLiMs necessary for optimal µHD engagement, both FCHO1/2 (*Umasankar et al., 2012*; *Henne et al., 2007*) and EPS15 (*Wang et al., 2016b*) binding to PtdIns$(4,5)P_2$, and the interdomain linker of FCHO1/2 regulating AP-2 (*Umasankar et al., 2014*; *Hollopeter et al., 2014*) to rearrange the µ2 subunit to co-align PtdIns$(4,5)P_2$, YXXØ and [DE]XXXL [LI] cargo binding interaction surfaces at the planar plasma membrane (*Jackson et al., 2010*). The data are also consistent with these rim-localized pioneers being able to drive deposition of new, open AP-2 into the interior of the incipient lattice.

It has recently been demonstrated that large, long-lived clathrin assemblies in human embryonic stem cell-derived fibroblasts correlate with elevated expression levels of AP-2 (*Dambournet et al., 2018*). Strikingly, while this may be mechanistically related to the abnormal clathrin-coated structures in the genome-edited HeLa clone #64/1.E and clone #40 cells, due to an increase in the cytosolic AP-2 pool, the effect of EPS15 impoundment in clone #40 cells shown here indicates that even with high soluble levels of AP-2 and PtdIns$(4,5)P_2$, normally sized and distributed clathrin-coated structures on the plasma membrane require the activity of EPS15/R and FCHO1/2. It has also just been reported that conditionally hematopoietic compartment compound-null Eps15$^{-/-}$/Eps15L1$^{-/-}$ mice are anemic due to dysfunctional erythrocyte differentiation and maturation (*Milesi et al., 2019*). Remarkably, although the transferrin receptor is the prototypical AP-2-dependent YXXØ-signal-harboring transmembrane receptor (*Traub and Bonifacino, 2013*; *Collawn et al., 1990*), the Eps15$^{-/-}$/Eps15L1$^{-/-}$ double knock-out mice display strikingly abnormal accumulation of the transferrin receptor on the surface of the defective circulating red blood cell population (*Milesi et al., 2019*). This dysregulation of the efficient internalization of the transferrin receptor in the erythrocyte precursor cells (reticulocytes) expressing normal AP-2 is obviously reminiscent of the phenotype of HeLa clone #40 cells with Nb E_142-mediated sequestration of EPS15 shown here, and independently reiterates the upstream role of Eps15/R (and Fcho1/2) in the proper functioning of the AP-2 adaptor complex under normal conditions.

## The utility of Nb-based tools for endocytosis research

The rationale behind generating Nbs against clathrin coat components is that, because clathrin-mediated endocytosis is an essential process, there is likely a limit to the number of component genes that can be systematically disrupted before overall cell viability becomes compromised. As shown here, tailored Nb-based intrabodies with in-frame subcellular compartment targeting information can impose relocation/mislocalization of Nb antigens, and change the dynamics, spread and accessibility of the soluble pool of the target antigen (*Beghein et al., 2016*). Repurposed E3 ligases can impose non-physiological turnover of Nb target proteins as an alternative to RNAi and genome-editing techniques (*Moutel et al., 2016*). These engineered biological tools do not require normal catabolic turnover of the target protein, as in si/shRNA-mediated gene silencing. Instead, as soon as the Nb-based chimera titrates out the target, either by compartmental sequestration or ectopic E3-mediated proteasomal degradation, any morphological or functional effect can be examined. Although not explored here, two other potential methods to promote rapid proteasomal degradation of EPS15/R with the Nb E_142 intrabody are by appending a degron sequence directly to the Nb (*Zhao et al., 2018*) or through TRIM21-mediated ubiquitination by making a combinatorial antibody fusing an Fc domain to the V$_H$H domain and using the 'Trim-Away' approach (*Clift et al., 2017*).

There are some constraints to the approaches reported here. For one thing, because they depend on transient transfection, the absolute expression level within the transfected cell population varies widely. While this can reveal dose-dependent effects, it hampers bulk population-based studies, as does the overall transfection efficiency. It might also be useful to modify the subcellular sequestration approach to make it inducible, either with chemical (*Stanton et al., 2018*) or

optogenetic (*Benedetti et al., 2018*) dimerizers. Future investigations can explore whether a strategy of electroporation of purified recombinant Nb E-142 protein chimeras (*Clift et al., 2017*) or appending a cell-penetrating adduct to the Nb (*Herce et al., 2017*) can alter the kinetics of the phenotypic response. An important practical feature of the assortment of tools described here is that, because of the highly conserved nature of the Nb framework sequence and fold, other, different antigen selective Nbs can be quickly and simply substituted for Nb E_142 by standard cloning techniques.

Finally, it is atypical that the NPF-harboring Nbs, especially Nb E_142, do not map to a single epitope upon the EH domain but rather present an orthosteric pseudoligand. This is a manifestation of the important structural differences between single-domain ($V_HH$) and classical dimeric heavy-light chain antibodies (*Dmitriev et al., 2016*; *Zimmermann et al., 2018*). Protruding CDR loops project off one edge of $V_HH$ Ig β-sandwich domain (*Figure 1E*), making them conformationally biased and poorly anti-peptide selective, quite unlike traditional heavy + light chain antibodies (*De Genst et al., 2006*; *Muyldermans et al., 2009*). This is the underlying reason for Nbs being able to recognize and stabilize different conformations of GPCRs and attendant signaling components (*Manglik et al., 2017*).

# Materials and methods

**Key resources table**

| Reagent type (species) or resource | Designation | Source or reference | Identifiers | Additional information |
|---|---|---|---|---|
| Gene (*Homo sapiens*) | EPS15 | UniProt | P42566 | Epidermal growth factor receptor substrate 15 |
| Gene (*H. sapiens*) | EPS15L1;EPS15R | UniProt | Q9UBC2 | Epidermal growth factor receptor substrate 15R |
| Gene (*H. sapiens*) | ITSN1 | UniProt | Q15811 | Intersectin-1 |
| Gene (*H. sapiens*) | FCHO1; KIAA0290 | UniProt | O14526 | FCH domain only protein 1 |
| Gene (*H. sapiens*) | FCHO2 | UniProt | Q0JRZ9 | FCH domain only protein 2 |
| Strain, strain background (*Escherichia coli*) | XL1-Blue | Agilent Technologies | #200236 | |
| Strain, strain background (*E. coli*) | BL21 Codon Plus RIL | Agilent Technologies | #230240 | |
| Strain, strain background (*E. coli*) | BL21 (DE3) Codon Plus RIPL | Agilent Technologies | #230280 | |
| Cell line (*H. sapiens*) (female) | HeLa SS6 | *Elbashir et al., 2001* | | Human cervical carcinoma cell line |
| Cell line (*H. sapiens*) (female) | HeLa SS6 clone #64/1.E | *Umasankar et al., 2014* | | Dual FCHO1- and FCHO2-gene disrupted (null) HeLa SS6 cells |
| Cell line (*H. sapiens*) (female) | HeLa SS6 clone #64/1.E #40 (clone #40) | this paper | | Triple FCHO1, FCHO2- and EPS15L1-gene disrupted (null) HeLa SS6 cells |
| Antibody | anti-β-actin; mouse clone mAb C4 | Santa Cruz Biotechnology | sc-47779 | 1:10,000 for immunoblotting |

*Continued on next page*

*Continued*

| Reagent type (species) or resource | Designation | Source or reference | Identifiers | Additional information |
|---|---|---|---|---|
| Antibody | anti-AP-2 α subunit mouse mAb C-8 | Santa Cruz Biotechnology | sc-17771 | 1:500 for immunoblotting |
| Antibody | anti-AP-2 α subunit; mouse clone mAb AP.6 | F Brodsky Laboratory | PMID: 2574457 | 1:1,000-1:2,000 for immunofluorescence |
| Antibody | anti-ARH rabbit antigen affinity purified pAb | L Traub Laboratory | PMID: 12451172 | 1:2,000 for immunoblotting |
| Antibody | anti-AP-1/2 β1/β2 subunit; rabbit antigen affinity purified pAb GD/1 | L Traub Laboratory | PMID: 7876268 | 1:2,500 for immunoblotting |
| Antibody | anti-CALM; goat affinity purified pAb C-18 | Santa Cruz Biotechnology | sc-6433 | 1:500 for immunoblotting |
| Antibody | anti-clathrin heavy chain; mouse clone mAb TD.1 | F Brodsky Laboratory | PMID: 1547490 | 1:5,000 for immunoblotting |
| Antibody | anti-Dab2 rabbit antigen affinity purified pAb | L Traub Laboratory | PMID: 12234931 | 1:2,500 for immunoblotting |
| Antibody | anti-Disabled-2/p96; mouse mAb 52/p96 | BD Transduction Laboratories | 610464 | 1:1,000 for immunoblotting |
| Antibody | anti-Eps15; rabbit affinity purified pAb C-20 | Santa Cruz Biotechnology | sc-534 | 1:500 for immunoblotting |
| Antibody | anti-Eps15; rabbit antigen affinity purified pAb | E Ungewickell Laboratory | PMID: 12960147 | 1:500-1:2,000 for immunofluorescence |
| Antibody | anti-Eps15R; rabbit mAb ab53006 | AbCAM | EP1147Y | 1:5,000 for immunoblotting |
| Antibody | anti-epsin 1; rabbit antigen affinity piurified pAb | L Traub Laboratory | PMID: 10692452 | 1:1,000 for immunoblotting |
| Antibody | anti-FCHO1; rabbit antigen affinity purified pAb | L Traub Laboratory | | 1:2,500 for immunoblotting |
| Antibody | anti-FCHO2; rabbit antigen affinity purified pAb | Novus Biologicals | NBP2-32694 | 1:700 for immunoblotting |
| Antibody | anti-GAPDH; mouse mAb | USP Bio | Y1041 | 1:3,500 for immunoblotting |
| Antibody | anti-GFP; antigen affinity purified pAb | P Hanson Laboratory | | 1:1,000 for immunoblotting |
| Antibody | anti-GFP-Booster; llama $V_H$H Nb | Chromotek | gba488-100 | 1:1,000-1:2000 for immunofluorescence |
| Antibody | anti-Hrb;RIP;Rab; goat affinity purified pAb | Santa Cruz Biotechnology | sc-1424 | 1:1,000 for immunoblotting |
| Antibody | anti-Hrb; AGFG1; mouse clone mAb H-2 | Santa Cruz Biotechnology | sc-166651 | 1:1,000-1:2,000 for immunofluorescence |
| Antibody | anti-intersectin 1; mouse clone mAb 29 | BD Transduction Laboratories | 611574 | 1:250 for immunoblotting |

*Continued*

| Reagent type (species) or resource | Designation | Source or reference | Identifiers | Additional information |
|---|---|---|---|---|
| Antibody | anti-LAMP-1; mouse clone mAb 1D4B | Developmental Studies Hybridoma Bank | AB_528127 | 1:1,000-1:2,000 for immunofluorescence |
| Antibody | anti-mNG; mouse clone mAb 32F6 | Chromotek | 32f6-100 | 1:1,000 for immunoblotting |
| Sequence - based reagent | pGEX-5X-1 | Pharmacia | | |
| Sequence- based reagent | pGEX-4T-1 | Pharmacia | | |
| Sequence - based reagent | pMW172 | Michael Way | | |
| Sequence - based reagent | pMECS | NSF/VIB | | |
| Sequence- based reagent | pHEN6c | NSF/VIB | | |
| Sequence - based reagent | pmNG-C1 | this paper | | mNG coding region replacing GFP in pEGFP-C1 vector |
| Sequence- based reagent | pEGFP-FYVE2 | this paper | | insertion of Nb E_142 following FYVE2 tandem |
| Sequence- based reagent | pEGFP-FYVE2-Nb E_142 | this paper | | introduction of 3maj destabilizing mutations into Nb E_142 |
| Sequence - based reagent | pEGFP-FYVE2-Nb dE_142 | this paper | | |
| Sequence - based reagent | piRFP-N1-Akap1-FRB | | | |
| Sequence - based reagent | piRFP-N1-Akap1-Nb E_142 | this paper | | replacement of control FRB module with Nb E_142 |
| Sequence - based reagent | pEGFP-FCHO1 (1–609) | *Umasankar et al., 2014* | | |
| Sequence- based reagent | PCR primer sets | this paper | | see *Table 1* for complete PCR primer set details |
| Peptide, recombinant protein | bovine serum albumin (BSA) | Pierce | 23209 | |
| Peptide, recombinant protein | glutathione | Sigma-Aldrich | G4251-1G | |
| Commercial assay or kit | QuikChange site-directed mutageneisis kit | Agilent Technologies | 200523 | |
| Chemical compound, drug | Glutathione-Spharose 4B | GE Healthscience | 17075605 | |
| Chemical compound, drug | Ni-NTA-agarose | MC Labs | NINTA-300 | |
| Chemical compound, drug | Normal goat serum | Sigma-Aldrich | 46767 | |
| Chemical compound, drug | Alexa Fluor 568-transferrin (*H. sapiens*) | Invitrogen; ThermoFisher Scientific | T23365 | |
| Chemical compound, drug | Saponin | Sigma-Aldrich | S7900 | |

*Continued on next page*

*Continued*

| Reagent type (species) or resource | Designation | Source or reference | Identifiers | Additional information |
|---|---|---|---|---|
| Chemical compound, drug | Lipofectamine 2000 | Invitrogen; ThermoFisher Scientific | 11668027 | |
| Software, algorithm | FV1-ASW 4.1 software | Olympus | | |
| Software, algorithm | ImageJ | *Rueden et al., 2017* | | |
| Software, algorithm | Fiji | *Schindelin et al., 2012* | | |
| Software, algorithm | NIS Elements 5.11.01 analysis package | Nikon | | |

Generally, all experiments (which were not randomized) were repeated at least twice on different days in technical replicates, with qualitatively similar results, or in biological replicates with little variation using different protein preparations, thawed cell passages and/or DNA preparations. Predetermination of sample size with statistical methods was not applied, and since data selection was not inherently subjective, no investigator blinding was used either during the experiments or for the evaluation of resulting data.

## Recombinant DNA

A plasmid encoding full-length (amino acids 1–896) *Hs* EPS15 in pGEX-5X-1 was kindly provided by Tomas Kirchhausen. The EH domain module (1-314) fused to the C-terminus of GST was generated from this parental plasmid using QuikChange site-directed mutagenesis (Agilent) to introduce a stop codon at position 315 (S315*) using the primers listed in *Table 1*. This plasmid was then used to make GST-EPS15 (1-109) and (1-217) by similar introduction of L110* and W218* stop codons (*Table 1*) using QuikChange. The GST-EPS15 (121-314) construct was prepared using Phusion mutagenesis and the phosphorylated primer set listed in *Table 1*. GST-EPS15R (1-365) was generated by PCR using primers (*Table 1*) and an EPS15L1 cDNA (BC142716, Transomic Technologies) and cloned into pGEX-5X-1 between BamHI and XhoI sites. The pMW172 vector containing GST fused to *Hs* intersectin 1 l (1-312), inserted between Not I and EcoRI restriction sites (*Snetkov et al., 2016*), was generously provided by Michael Way.

The cDNAs encoding Nb E_3, E_142 and E_180 were amplified by PCR from preparations in pMECS using the primers A6E and PMCF (*Table 1*) and cloned into pHEN6c between PstI and BstEII sites. This encodes a Nb with an N-terminal 22-residue PelB sequence for periplasmic expression and a His$_6$ tag at the C terminus. QuikChange mutagenesis was used to introduce the NPF→APA substitution in the Nb E_142 pHEN6c plasmid using an appropriate primer set (*Table 1*).

Nb E_3/142/180 were amplified by PCR for cloning into pEGFP-N1 vector using universal primers (*Table 1*). For insertion of these Nbs into a mNG-encoding plasmid, first mNG was amplified from a commercially-synthesized mNG clone (Genscript) using a primer pair (*Table 1*) and then substituted for eGFP in the pEGFP-C1 vector using the AgeI and XhoI restriction sites. Nb E_3 and E_142 were then subcloned into this vector utilizing PCR with primers (*Table 1*) allowing in-frame insertion with the BglII and HindIII sites and adding four glycine residues between the mNG and the Nb. For construction of the in-frame tandem mNG-Nb E_142 × 3 vector, the cloning involved sequential addition of the PCR-amplified (primer pairs reported in *Table 1*) Nb E-142 cDNA between HindIII and SalI sites with a (Ser-Gly-Gly)$_3$ spacer between Nb repeat 1 and 2, insertion of a third Nb E_142 cDNA between SalI and BamHI sites with a Val-Asp-Gly-Gly-Ser-Gly-Gly-Ser-Gly-Gly spacer between Nb 2 and 3 and a 3' TAA stop codon and, finally, directional cloning replacement of the first Nb E_142 repeat between BglII and HindIII sites to remove the original in-frame stop codon.

GFP-FYVE$_2$-*Hs* FCHO1 µHD plasmid, modified from the GFP-FYVE$_2$-Ub$^{\Delta GG}$ plasmid from Pietro de Camilli (*Chen and De Camilli, 2005*), has been described previously (*Ma et al., 2016*). From this, the control plasmid GFP-FYVE$_2$ was made using primers to introduce a stop codon (E410*) by

**Table 1.** PCR primer sets.

| Primer pair | Plasmid | Method | FORWARD 5′ sequence | REVERSE 3′ sequence |
|---|---|---|---|---|
| HsEPS15 315->*FORWARD QC/HSEPS15 315->*REVERSE QC | Hs EPS15 (1-314) pGEX-5X-1 | QuikChange mutagenesis | 5′-ATGATTCCACCATCAGACAGGGCCTGATTACAAAAGAACATCATAGG-3′ | 5′-CCTATGATGTTCTTTTGTAATCAGGCCCTGTCTGATGGTGGAATCAT-3′ |
| EH2 W218* QC FORWARD/EH2 W218* QC REVERSE | Hs EPS15 (1-217) pGEX-5X-1 | QuikChange mutagenesis | 5′-CCATCTAAGAGAAAAACGTGAGTTGTATCCCCTGCAG-3′ | 5′-CTGCAGGGGATACAACTCACGTTTTTCTCTTAGATGG-3′ |
| EH1 L110* QC FORWARD/EH1 L110* QC REVERSE | Hs EPS15 (1-109) pGEX-5X-1 | QuikChange mutagenesis | 5′-CCAAGATTTCATGATACCAGTAGTCCTTAACTAATCAGTGGAACCTCTGC-3′ | 5′-GCAGAGGTTCCACTGATTAGTTAAGGACTACTGGTATCATGAAATCTTGG-3′ |
| Hs EPS15 Pro121 Phusion FORWARD/pGEX-5X-1 BamHI Phusion REVERSE | Hs EPS15 (121-314) pGEX-4T-1 | Phusion mutagenesis | 5′-phos-CCATGGGCTGTAAAACCTGAAGATAAGG-3′ | 5′-phos-GATCCCACGACCTTCGATCAGATC-3′ |
| EPS15 W57A QC FORWARD/EPS15 W57A QC REVERSE | Hs EPS15 (1–107/W57A) pGEX-5X-1 | QuikChange mutagenesis | 5′-GACTTGATACTTGGAAAGATTGCGGATTTAGCCGACACAGATGGC-3′ | 5′-GCCATCTGTGTCGGCTAAATCCGCAATCTTTCCAAGTATCAAGTC-3′ |
| Hs EPS15L1 EH1 BamHI-Met1 FORWARD/Hs EPS15L1 EH3 T366* XhoI REVERSE | Hs EPS15R (1-365) pGEX-5X-1 | Directional cloning | 5′-GTG*GGATCC*TCACCATGGCGGCGCCGCTCATCC-3′ | 5′-ATA*CTCGAG*TTAGCCTCTCTCCGAAGGCGGGACCATGTCC−3′ |
| Primer A6E/Primer PCMF | Nb E_3-His6 pHEN6c | Directional cloning | 5′-GATGTGCAG*CTGCAG*GAGTCTGGRGGAGG-3′ | 5′-CTAGT*GCGGCCGC*TGAGGAGAC*GGTGACC*TGGGT-3′ |
| Primer A6E/Primer PCMF | Nb E-_142-His6 pHEN6c | Directional cloning | 5′-GATGTGCAG*CTGCAG*GAGTCTGGRGGAGG-3′ | 5′-CTAGT*GCGGCCGC*TGAGGAGAC*GGTGACC*TGGGT-3′ |
| Primer A6E/Primer PCMF | Nb E_180-His6 pHEN6c | Directional cloning | 5′-GATGTGCAG*CTGCAG*GAGTCTGGRGGAGG-3′ | 5′-CTAGT*GCGGCCGC*TGAGGAGAC*GGTGACC*TGGGT-3′ |
| NPF-APA QC primer FORWARD/NPF-APA QC primer REVERSE | Nb E-_142 (NPF APA)-His6 pHEN6c | QuikChange mutagenesis | 5′-TGTAATATGAAGAGCATGGCCCCCGCCAGACGGTATGACGTGTGG-3′ | 5′-CCACACGTCATACCGTCTGGCGGGGGCCATGCTCTTCATATTACA-3′ |
| AgeI mNeonGreen 5′ FORWARD/XhoI mNeonGreen 3′ REVERSE | mNG pmNG-C1 | Directional cloning | 5′-GCGCTACCGGTCGCCACCATGGTGAGCAAGGGCGAGG AGG-3′ | 5′-TTGAGCTCGAGATCTGAGTCCGGACTTGTACAGCTCGTCCATGCCCATCAC-3′ |
| Nb 5′ BglII + Gly4 FORWARD/Nb 3′ HindIII REVERSE | mNG-Nb E_3 pmNG-C1 | Directional cloning | 5′-ATC*AGATCT*GGAGGTGGAGGTCAGGTGCAGCTGCAGG AG-3′ | 5′-CTA*AAGCTT*TTATGAGGAGACGGTGACCTGGG-3′ |
| Nb 5′ BglII + Gly4 FORWARD/Nb 3′ HindIII REVERSE | mNG-Nb E_142 pmNG-C1 | Directional cloning | 5′-ATC*AGATCT*GGAGGTGGAGGTCAGGTGCAGCTGCAGG AG-3′ | 5′-CTA*AAGCTT*TTATGAGGAGACGGTGACCTGGG-3′ |
| CORRECT_3 × 1–2 Nb E_142 BmsBI + Gly6Ser2 FORWARD/ | mNG-Nb E_142 × 3 pmNG-C1 | Directional cloning | 5′-CACCGTCTCCTCATCCGGAGGTTCAGGAGGTTCCGGGGGCCAGGTGCAGCTGCAGG AG-3′ | 5′-GCCGTCGACTGAGGAGACGGTGACCTG-3′ |
| 3 × 2–3 Nb E_142 SalI + Gly6Ser2 FORWARD/BamHI 3x EPS142 Nb REVERSE for mNG-C1 | mNG-Nb E_142 × 3 pmNG-C1 | Directional cloning | 5′-CTCAGTCGACGGCGGATCTGGTGGGAGTGGGGGACAGGTGCAGCTGCAGGAGTC-3′ | 5′-GGTGGATCCTTATGAGGAGACGGTGACCTG-3′ |
| Nb 5′ BglII + Gly4 FORWARD/2X Nb E_142 QC primer HindIII-SalI REVERSE | mNG-Nb E_142 × 3 pmNG-C1 | Directional cloning | 5′-ATC*AGATCT*GGAGGTGGAGGTCAGGTGCAGCTGCAGG AG-3′ | 5′-CCGCCGTCGACTGCACGAAATTCGAAGCTTGAGGAGACGGTGACCTGGGT-3′ |
| QC GFP-FYVE2-UbDelGG EcoRI sto/QC GFP-FYVE2-UbDelGG EcoRI sto | GFP-FYVE2 pEGFP-C1 | QuikChange mutagenesis | 5′-GAGCAGCTGAACAAGAAGCATAATTCATGCAGATCTTT GTG-3′ | 5′-CACAAAGATCTGCATGAATTATGCCTTCTTGTTCAGCTGCTC-3′ |
| VHH Nb EcoRI +Gly4 forward/Anti-Eps15 VHH Nb ApaI reverse | GFP-FYVE2-Nb E_142 pEGFP-C1 | Directional cloning | 5′-ATC*GAATTC*GGAGGTGGAGGTCAGGTGCAGCTGCA GGAG-3′ | 5′-ACT*GGGCCC*TTATGAGGAGACGGTGACCTG-3′ |
| VHH Nb EcoRI +Gly4 forward/3 maj Anti-Eps15 VHH Nb ApaI reverse | GFP-FYVE2-dNb E_142 pEGFP-C1 | Directional cloning | 5′-ATC*GAATTC*GGAGGTGGAGGTCAGGTGCAGCTGCAGG AG-3′ | 5′-ACT*GGGCCC*TTAGAAGGAGACGGTGACCTG-3′ |
| Nb + NLS C-term Phusion FORWARD/Nb C- term Phusion REVERSE | GFP-Nb E_142-SV40 NLS pEGFP-C1 | Phusion mutagenesis | 5′-phos-GGACCCAAGAAGAAACGGAAGGTGTGAGGGCCCGGGATCCACCG-3′ | 5′-phos-GCGGCCGCCTGAGGAGACGGTGACCTGGGTCC-3′ |

*Table 1 continued on next page*

*Table 1 continued*

| Primer pair | Plasmid | Method | FORWARD 5' sequence | REVERSE 3' sequence |
|---|---|---|---|---|
| MfeI EPS Nb142 FORWARD for GNbE3/BglII EPS15 Nb142 REVERSE for GNbE3 | G-Nb E_142-E3 pCAG_EGFP | Directional cloning | 5'-TCGAG*CAATTG*ATGGCAGA AGTTCAGGTGCAGCTGCAGG AG-3' | 5'-GGTA*AGATCT*TCCTGAGGAG ACGGTGACCTGGGTCCCCTGG CCC-3' |

DOI: https://doi.org/10.7554/eLife.41768.021

QuikChange mutagenesis (*Table 1*). Nb E_3, E_142 and E_180 were amplified by PCR with primers (*Table 1*) and substituted for the FCHO1 µHD between EcoRI and ApaI sites, adding four glycine residues following the second tandem FYVE domain. The conditionally stable version of Nb E_142, with the 3maj substitutions (S73R, C98Y and S117F; *Tang et al., 2016*) was synthesized commercially (IDT) and subcloned into the GFP-FYVE$_2$-plasmid similarly using a primer pair (*Table 1*) and the same restriction sites. The Akap1-FRB-iRFP-encoding plasmid (piRFP-N1-Akap1-FRB) was provided by Gerry Hammond. Using primers (*Table 1*) Nb E_142 was PCR amplified and substituted for the FRB module between BglII and AgeI sites, adding a Gly-Ser-Gly-Ser-Gly linker at the 3' end. The GFP-PLCδ1 PH domain PtdIns(4,5)P$_2$ sensor and construction of the GFP-Hs FCHO1 (1–609) and (1-467) plasmids have been described (*Umasankar et al., 2014*). pCAG_EGFP-GFE3 (plasmid #79869) (*Gross et al., 2016*) was obtained from Addgene. To replace the central FingR domain with Nb E_142, PCR using primers that allowed cloning of the Nb between MfeI and BglII sites (*Table 1*) was used.

All plasmid constructs were verified by automated double-stranded dideoxynucleotide sequencing (Genewiz). Full sequence details and maps are available upon request.

## Recombinant protein purification

GST fusion proteins were synthesized in the *E. coli* BL21 codon plus RIL strain (Invitrogen). After transformation, GST and all GST-fusion proteins were purified by a standardized protocol involving induction of cultures grown in 2 x YT medium at 37°C to an OD$_{600}$ of ~0.6–0.8 with 100 µM IPTG at 18°C overnight. Bacteria were then collected by centrifugation (15,300 x $g_{max}$, 4°C, 10 min) and pellets resuspended in a 3:1 mixture of B-PER and Y-PER (Thermo-Fisher) with 0.25 mg/ml freshly prepared hen egg lysozyme and 50 U/ml benzonase nuclease (Sigma). During resuspension, PMSF was added to 1 mM from a 200 mM stock solution in isopropanol. The bacterial suspension was centrifuged (27,000 x $g_{max}$, 4°C, 15 min) and then the supernatant fraction collected and incubated batch wise at 4°C with continuous up-and-down mixing of 0.5–2.0 ml added glutathione-Sepharose (GE), as required. After initial washing of the Sepharose with ice-cold PBS by centrifugation, the beads were transferred to a disposable polypropylene column on ice and washed with ~10 column volumes of PBS. Bound GST/GST-fusion proteins were eluted with 25 mM Tris-HCl, 250 mM sodium chloride, 10 mM reduced glutathione and 2 mM DTT on ice. Peak fractions were pooled and dialyzed into PBS at 4°C. Protein concentration was determined using the Bradford method with BSA as the standard and aliquots stored at −80°C.

## Nb production, screening and purification

For the generation of the Nb clones, immunization, library construction and screening was carried out by the Nanobody Service Facility at the Vrije Universiteit Brussel, Belgium. The entire procedure was as described in detail (*Vincke et al., 2012*). Briefly, a single llama was inoculated six times over a 35 day period with a mixture of 250 µg purified *Hs* EPS15 (1-314) and Gerbu adjuvant P. Purified lymphocytes from blood collected on day 39 were used to isolate mRNA for cDNA synthesis by RT-PCR with and oligo(dT) primer. The immune nanobody library construction utilized this cDNA and appropriate primers to amplify V$_H$H antigen specific sequences. PCR products were digested with PstI and NotI for subcloning into the phagemid vector pMECS. From the library of ~3×10$^8$ transformants, 97% (93 of 95 clones) had correctly sized inserts and sequencing confirmed the presence of germ line V$_H$H sequences. Antigen-specific clones were identified by solid phase biopanning *E. coli* TG1 infected with the library using 200 µg/ml of the immunizing EPS15 (1-314) immobilized on plates. After two sequential rounds of selection of phagemid particles, 14 specific clones were obtained, which sequencing revealed corresponded to seven individual V$_H$H sequences.

Small-scale periplasmic extracts were prepared from 60 ml overnight cultures of *E. coli* XL1-Blue transformed with the appropriate Nb clone in pHEN6c (*Vincke et al., 2012*). Bacteria were initially grown at 37°C to an $OD_{600}$ of ~0.8–1.0 in LB supplemented with 0.1% glucose, 2 mM magnesium chloride and ampicillin and then IPTG added to a final concentration of 1 mM before incubation overnight at 28°C. Large-scale inductions were carried out similarly but using 500 ml cultures. Bacteria were collected by centrifugation (9000 x $g_{max}$, 4°C, 15 min) and resuspended in ice-cold TES buffer (200 mM Tris-HCl, pH 8.0, 0.5 mM EDTA, 500 mM sucrose). The bacterial suspension was incubated at 4°C with continual vigorous mixing at 250 rpm on an orbital shaker for ~60 min before addition of 2 volumes of 1/4-strength TES (50 mM Tris-HCl, pH 8.0, 0.125 mM EDTA, 125 mM sucrose). After an additional 60–90 min incubation at 4°C with continual mixing at 250 rpm, the bacteria were pelleted by centrifugation at 10,000 × $g_{max}$ at 4°C for 30 min and the supernatant (periplasmic extract) aspirated and stored in aliquots at −80°C. Before use in biochemical binding assays, samples of periplasmic extract were adjusted to 125 mM sodium chloride by addition of 1/40 vol of a 5 M stock solution on ice and then centrifuged at ~125,000 x $g_{max}$ (45,000 rpm, Beckman TLA 55 rotor) at 4°C for 20 min. Aliquots of the clarified supernants were added to immobilized GST/GST-fusion proteins bound to washed glutathione-Sepharose beads for pull-down assays.

To purify the Nb E_142-His$_6$ protein from periplasmic extract, the protein was first purified over a 1.5 mL column Ni-NTA-agarose (MC Labs), eluted with 500 mM imidazole, concentrated to 5 ml using an Amicon Ultra 15 device with a 3000 nominal molecular weight limit and then 2.5 ml fractionated at 0.8 ml/min in the upward direction over a HiLoad Sephacryl S-100 HR column (16 × 60 mm, GE) equilibrated in 50 mM Tris-HCl, pH 7.6, 150 mM sodium chloride. Peak 2 ml fractions identified by SDS-PAGE were pooled, and concentrated with an Amicon Ultra device to ~4 ml with a protein concentration of ~1.75 mg/ml. Aliquots of the purified Nb E_142 were stored at −80°C.

For biochemical interaction assays, first, purified GST or the appropriate GST-fusion protein (25–200 µg) was immobilized on 25 µl of packed glutathione-Sepharose (GE) by incubation in 750 µl PBS at 4°C for ~2 hr with continual mixing. The beads were then recovered by centrifugation at 10,000 x $g_{max}$ for 1 min and washed into either PBS or assay buffer (25 mM Hepes-KOH, pH 7.2, 125 mM potassium acetate, 5 mM magnesium acetate, 2 mM EDTA, 2 mM EGTA and 2 mM DTT) and resuspended to a total volume of 50 µl in residual wash buffer. Clarified periplasmic extract or HeLa cell Triton X-100 lysate was added to the washed beads and incubation continued at 4°C for 60 min with continuous up-and-down mixing. The beads were again sedimented by centrifugation (10,000 x $g_{max}$, 1 min) and a 70 µl aliquot of each supernatant fraction removed and transferred to a new microfuge tube on ice. After aspirating the remainder of each supernatant, the pellets were washed 4 x with 1.5 ml/wash ice-cold PBS and then the Sepharose bead pellet resuspended in SDS sample buffer to a final volume of 100 µl. After briefly boiling, aliquots of 2% of each supernatant and 10% of each supernatant were resolved by SDS-PAGE and either stained with Coomassie brilliant blue in 40% methanol, 10% acetic acid or transferred to nitrocellulose in a cold buffer of 15.6 mM Tris, 120 mM glycine at 110V for 75 min. After staining the blots with 0.1% (w/v) Ponceau S in 5% acetic acid (Sigma) to identify and mark the molecular mass standards, the nitrocellulose was cut into appropriate replicate sections, destained in transfer buffer, and blocked overnight in 5% non-fat milk powder in TBST (25 mM Tris-HCl, pH 7.5, 150 mM sodium chloride, 0.1% Tween 20) at room temperature. Quantitative binding assays (*Pollard, 2010*) were performed at 4°C for 60 min in PBS containing 100 µg/ml carrier BSA. The final assay volume was 400 µl, to allow calculation of molarities using the molecular weights of 13,683.2 for Nb E_142-His$_6$, 26,200 for GST, 38,271.3 for GST-Hs EPS15 (1–109/EH1) and 60,924.7 for GST-Hs EPS15 (1–314/EH1-3). The Nb E_142 concentration was fixed at 2 µM and GST-Hs EPS15 (1-109) varied from 0.5 to 20 µM while (1-314) was titrated over the range 0.25–10 µM. The intensity of the Nb E_142 band in the supernatant fractions, collected under equilibrium conditions (*Pollard, 2010*), was determined with ImageJ/Fiji (*Schindelin et al., 2012*; *Rueden et al., 2017*) and data normalized to the Nb E_142 found in the GST control supernatant fraction.

A complete list of the primary antibodies used in this study are provided in *Table 2*. Primary antibodies were diluted in 1% non-fat milk in TBST and typically incubated with the nitrocellulose-immobilized protein antigens at room temperature for 2 hr. After thorough washing with TBST, appropriate HRP-conjugated anti-mouse/rabbit/goat antibodies (Amersham/GE Healthcare) diluted in 1% milk in TBST were added and incubated at room temperature for 60 min. Following extensive

washing in TBST, blots were briefly incubated with ECL-type reagent mixture (HyGLOW, Denville) and signals collected on X-ray film.

## Cells lines and tissue culture

The parental cell line HeLa SS6 (*Elbashir et al., 2001*), derived from the female cervical adenocarcinoma HeLa line, has the characteristic compact, dense epithelial morphology and was verified of *H. sapiens* origin by sequencing results (*Umasankar et al., 2014*). Cell line authentication using autosomal short tandem repeats (STRs) by the University of Arizona Genetics Core coincides with HeLa above an 80% match threshold. The HeLa SS clone #64/1.E (clone 1.E cells) were produced previously by sequential TALEN-mediated gene disruption of the *FCHO2* locus on chromosome five and the *FCHO1* locus on chromosome 19 (*Umasankar et al., 2014*). TALEN-mediated generation of clones #15, #19 and #40 from the #64/1.E line was as we have described; assembly of the TALEN arrays (*Joung and Sander, 2013*), transfection and selection were by the procedures outlined (*Umasankar et al., 2014*). All HeLa cells were cultured in Dulbecco's modified Eagle's medium (DMEM, Lonza) supplemented with 10% FCS (Atlanta Biologicals) and 2 mM L-glutamine (Gibco) in a humidified 5% $CO_2$ atmosphere at 37°C. K-652 cells were grown in RPMI 1640 medium supplemented with 5% FCS and 2 mM L-glutamine at 37°C. K-562 cells, derived from a female pleural effusion (*Lozzio and Lozzio, 1975*), were not STR authenticated, but grow in suspension culture as

**Table 2.** Primary antibodies.

| Antigen | Species | Designation/clone | Catalogue number | Supplier/Source | Application |
|---|---|---|---|---|---|
| β-actin | mouse | mAb C4 | sc-47778 | Santa Cruz Biotechnology | immunoblot |
| AP-2 α subunit | mouse | mAb C-8 | sc-17771 | Santa Cruz Biotechnology | immunoblot |
| AP-2 α subunit | mouse | mAb AP.6 | — | Frances Brodsky | immunofluorescence |
| ARH | rabbit | antigen affinity purified pAb | — | Traub laboratory | immunoblot |
| AP-1/2 β1/β2 subunit | rabbit | antigen affinity purified pAb GD/1 | — | Traub laboratory | immunoblot |
| CALM | goat | affinity purified pAb C-18 | sc-6433 | Santa Cruz Biotechnology | |
| clathrin heavy chain | mouse | mAb TD.1 | — | Frances Brodsky | immunoblot |
| Dab2 | rabbit | antigen affinity purified pAb | — | Traub laboratory | immunoblot |
| Disabled-2/p96 | mouse | mAb 52/p96 | 610464 | BD Transduction Laboratories | immunoblot |
| Eps15 | rabbit | affinity purified pAb C-20 | sc-534 | Santa Cruz Biotechnology | immunoblot |
| Eps15 | rabbit | antigen affinity purified pAb | — | Ernst Ungewickell | immunofluorescence |
| Eps15R | rabbit | mAb ab53006 | EP1147Y | AbCAM | immunoblot |
| epsin 1 | rabbit | antigen affinity purified pAb | — | Traub laboratory | immunoblot |
| FCHO1 | rabbit | antigen affinity purified pAb | — | Traub laboratory | immunoblot |
| FCHO2 | rabbit | antigen affinity purified pAb | NBP2-32694 | Novus Biologicals | immunoblot |
| GAPDH | mouse | mAb | Y1041 | UBP Bio | immunoblot |
| GFP | rabbit | antigen affinity purified pAb | — | Phyllis Hanson | immunoblot |
| GFP-Booster | llama | anti-GFP Nb Atto 488 conjugate | gba488-100 | Chromotek | immunofluorescence |
| Hrb/RIP/Rab | goat | affinity purified pAb C-19 | sc-1424 | Santa Cruz Biotechnology | immunoblot |
| Hrb/AGFG1 | mouse | mAb H-2 | sc-166651 | Santa Cruz Biotechnology | immunofluorescence |
| Intersectin 1 | mouse | mAb clone 29 | 611574 | BD Transduction Laboratories | immunoblot |
| LAMP-1 | mouse | mAb clone 1D4B | AB_528127 | Developmental Studies Hybridoma Bank | immunoblot |
| mNG | mouse | mAb clone 32F6 | 32f6-100 | Chromotek | immunoblot |

DOI: https://doi.org/10.7554/eLife.41768.022

expected, react with human-specific antibodies, and express both FCHO1 and FCHO2 proteins (*Umasankar et al., 2014*) and EPS15 and EPS15R, as predicted by mRNA expression profiling (*Uhlén et al., 2015*). Cells tested negative for mycoplasma using a PCR-based assay, and were routinely discarded after no more than 30 passages.

Triton X-100 whole cell lysates were prepared from parental HeLa SS6 cells cultured in at least $5 \times 15$ cm Petri dishes and collected into CellStripper dissociation reagent (Corning).

For transfections, Lipofectamine 2000 (Invitrogen) was used. First, media from the cells plated in 35 mm culture dishes or 6-well trays was aspirated and replenished with 2 ml of fresh DMEM, 10% FCS, 2 mM glutamine. Aliquots of the appropriate endotoxin-free mini/midiprep plasmid DNA (0.4– 1.0 µg) diluted into 375 µl Opti-MEM (Gibco) were incubated at room temperature for ~15 min with occasional mixing before addition of 5 µl of the Lipofectamine 2000 stock solution to each tube of DNA. After incubation at room temperature for an additional ~45 min, the transfection mixture was added drop wise to the cells and incubated at 37°C overnight in a humidified atmosphere with 5% $CO_2$. After 16–20 hr, cells were used for endocytosis assays or either fixed for morphological analysis or processed for immunoblotting. Cells were detached by incubation at room temperature with Cell-Stripper and collected into microfuge tubes on ice. After rinsing the dishes with PBS and pooling with the CellStripper suspension, cells were recovered by centrifugation ($500 \times g_{max}$ for 5 min at 4°C) and the supernatants aspirated and discarded. Each pellet was resuspended in hot SDS sample buffer and boiled before analysis by SDS-PAGE.

For transferrin uptake assays, cells on cover slips were first incubated in a starvation medium composed of DMEM, 25 mM Hepes, pH 7.2, 0.5% BSA and 2 mM L-glutamine, for 60 min at 37°C in 5% $CO_2$ to discharge intact apotransferrin from surface transferrin receptors. For the 5 min pulse experiments, cells were incubated in 500 µl starvation medium supplemented with 25 µg/ml Alexa Fluor 568-conjugated human transferrin (Life Technologies/Thermo Fisher) at 37°C, then placed immediately into an ice/water bath and rapidly washed three times with ice-cold PBS before fixation in ice-cold 4% paraformaldehyde (Electron Microscopy Sciences) in PBS on ice for 10 min. For the modified uptake assay (*Reis et al., 2015*), starved HeLa clone #40 cells were incubated 25 µg/ml Alexa Fluor 568-transferrin in 500 µl at 37°C for 2 min, transferred to an ice/water bath and washed three times with 0.2 M acetic acid, 0.5 M sodium chloride, pH 2.5 (*Lamb et al., 1983*). Then 500 µl of warmed starvation medium was added to the cells and incubation continued at 37°C for a further 2 min before fixation in paraformadehyde on ice.

## Immunofluorescence analysis

After removing culture medium, cells attached to 12 mm round glass cover slips were fixed in ice-cold 4% paraformaldehyde in PBS on ice for 10 min. The cells were then washed three times with ice-cold PBS before incubation in 5% normal goat serum (Sigma), 0.2% saponin (Sigma) in PBS at room temperature for at least 30 min. For incubation with primary antibodies, appropriate dilutions were prepared in 5% normal goat serum, 0.2% saponin in PBS and centrifuged at $15,000 \times g_{max}$ for 10 min before addition to the fixed and permabilized cells. Incubation with primary antibodies was for 60 min at room temperature, followed by three washes with PBS. Fluorochrome-coupled secondary antibodies (Invitrogen) were similarly diluted (1:1,000) in 5% normal goat serum, 0.2% saponin in PBS and centrifuged at $15,000 \times g_{max}$ for 10 min before addition of aliquots of the clarified supernatant to the cells. For double transfection experiments involving both iRFP and GFP, where the efficiency of GFP expression is sharply reduced, an anti-GFP nanobody (GFP-Booster, Chromotek) conjugated to Atto 488 was used to detect GFP-expressing cells unambiguously. After incubation at room temperature for 30–45 min, the cells were washed three times with PBS, in some cases incubated briefly in 2 µg/ml Hoechst 33258 DNA dye in PBS, and mounted on glass slides in either Cytoseal (Thermo Fisher) or SlowFade Diamond (Life Technologies) mounting medium.

## Confocal microscopy and image processing

All images were acquired using an Olympus FV1000 confocal microscope equipped with a $60 \times$ PlanApo (NA 1.42) oil objective lens and controlled by the Olympus Fluoview FV1-ASW 4.1 software. Fluorochromes were excited with the laser bench combiner as follows: a 405 nm laser for Hoechst, the 488 nm laser for Alexa Fluor 488, Atto 488, GFP and mNG, the 559 nm laser for Alexa Fluor 546 and −567, and the 635 nm laser for Alexa Fluor 647. The $1,024 \times 1,024$ pixel images, with

12-bit depth per color channel, were always acquired by sequential scanning with line-based Kalman averaging filter mode and saved as TIFF files. Sequential z-stacks were collected with a 0.3 µm-step size between sections. Minor adjustments to intensity and contrast were performed in Adobe Photoshop CC and composite images were assembled and annotated in Adobe Illustrator CC.

Quantitation of fluorescence confocal images was similar to that described previously (*Ma et al., 2016*), but performed using the Nikon NIS Elements 5.11.01 analysis package. In some cases, confocal z-stack images were first deconvoluted and assembled into a single maximum-projection image as previously described (*Ma et al., 2016*). For all images, after threshold-based background correction of raw files, a binary mask of objects was generated by processing for punctate structures (using the Advanced Denoising and Detect Regional Maxima function), and then individual statistics for object size computed. Fluorescence intensity measurements were determined by analyzing full 3D volumes, again by using the Denoising function, smoothing, deconvolution, and 3D binary thresholding and segmentation in NIS Elements. The original transferrin channel was then quantified using the process-generated mask. In the 1,024 × 1,024 pixel images acquired with the FV1000 60 × objective and 2 × magnification, the individual pixel size is 0.103 nm. For comparisons in the transiently-transfected cell image data sets, region-of-interest segmentation was performed on the basis of the distribution of the fluorescence signal in transfected cell marker channel. Counted objects were classified into area-based or volume-based category bins as described previously (*Ma et al., 2016*). In most instances, comparable total cell areas were compared, as stated in the individual figure legends. Usually, five separate data sets were analyzed, and the data combined for presentation in the figures. Dot/boxplots were generated using PlotsOfData (*Postma and Goedhart, 2019*).

## Acknowledgements

This manuscript is dedicated to Dr. Anastasios Raptis and his outstanding treatment team at the Mario Lemieux Center for Blood Cancers. Without their dedicated medical skill, clinical knowledge and compassionate care and support this work would never have been completed.

Gholamreza Hassanzadeh Ghassabeh and his team in Brussels, Belgium provided outstanding service in generating, screening and ELISA testing the Nb clones. I am grateful to P.K. Umasankar and Sachin Holkar for their efforts early in this project in generating the *EPS15L1* gene-edited HeLa cell lines. I am enormously grateful to Simon Watkins and Callen Wallace for graciously deconvolving confocal image stacks for me and for skillfully performing the quantitation of the immunofluorescence images. Tomas Kirchhausen, Michael Way, Peter Drain, Pietro de Camilli, Gerry Hammond and Donald Arnold supplied important plasmids. I am also extremely grateful to David Owen, who provided critically insightful comments on the manuscript.

This work was supported by NIH award R01GM106963.

## Additional information

### Funding

| Funder | Grant reference number | Author |
| --- | --- | --- |
| National Institute of General Medical Sciences | R01GM106963 | Linton M Traub |

The funders had no role in study design, data collection and interpretation, or the decision to submit the work for publication.

### Author contributions

Linton M Traub, Conceptualization, Resources, Data curation, Formal analysis, Funding acquisition, Validation, Investigation, Visualization, Methodology, Writing—original draft, Project administration, Writing—review and editing

### Author ORCIDs

Linton M Traub (ID) https://orcid.org/0000-0002-1303-0298

Decision letter and Author response
Decision letter https://doi.org/10.7554/eLife.41768.025
Author response https://doi.org/10.7554/eLife.41768.026

## Additional files

### Supplementary files
• Transparent reporting form
DOI: https://doi.org/10.7554/eLife.41768.023

### Data availability
All data generated or analysed during this study are included in the manuscript and supporting files. Key plasmids will be made available via Addgene.

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
