## [Decision Letter]

Thank you for submitting your Research Advance "A nanobody-based molecular toolkit provides new mechanistic insight into clathrin-coat initiation" for consideration by *eLife*. Your article has been favorably reviewed by three peer reviewers, including Suzanne Pfeffer as the Reviewing Editor and Reviewer #1, and the evaluation has been overseen by Randy Schekman as the Senior Editor.

The reviewers have discussed the reviews with one another and the Reviewing Editor has drafted this decision to help you prepare a revised submission.

Summary:

In this high quality manuscript, the author has generated nanobodies against the clathrin coat component, EPS15 to study coat assembly in cells. The author anchored a nanobody to early endosomes or mitochondria to modulate the levels of the protein that is available for coat assembly. One of the nanobodies is a pseudoligand that provides information about binding site occupancy in cells. He also studies new triple null cells and finds that without FCHO1/2 and EPS15/R, coated assemblies do not form stably, and pioneer proteins at the perimeter of clathrin coated structures are needed to form new buds. This paper presents a valuable tool and the reviewers support presentation of this story in *eLife* in the Tools and Resources category.

The reviewers agreed that these two points would greatly enhance the manuscript.

1) It remains unclear whether the nanobody is entirely specific to Eps15/Eps15R. It seems reasonable to test whether other related EH domain containing proteins or isolated EH domains (e.g. from intersectin) also bind to the nanobodies isolated. This is important to define, if the goal is to demonstrate a highly specific effect of depleting Eps15/Eps15R in cells. Is there any data available on the affinity of the nb for a single EH domain or for an array of them? A standard solution technique should sort this out, and if not too difficult, would add valuable information for this story.

2) The work would benefit greatly from at least some quantification of the images (Image J or cellprofiler) – for example, the amount of internalized transferrin can be quantified, and the amount of AP2 or other protein per square micron for example, can be quantified. Where a nanobody has an effect, can one correlate extent of phenotype as a function of level of nanobody? This is not meant to be onerous but should really not be too hard with available image software.

---

## [Author Response]

The reviewers agreed that these two points would greatly enhance the manuscript.1) It remains unclear whether the nanobody is entirely specific to Eps15/Eps15R. It seems reasonable to test whether other related EH domain containing proteins or isolated EH domains (e.g. from intersectin) also bind to the nanobodies isolated. This is important to define, if the goal is to demonstrate a highly specific effect of depleting Eps15/Eps15R in cells. Is there any data available on the affinity of the nb for a single EH domain or for an array of them? A standard solution technique should sort this out, and if not too difficult, would add valuable information for this story.

I fully concur that defining more coherently the selectivity of the newly-described nanobody reagents for EH domains is a critical point, especially for unambiguous interpretation of the cell-based results. Because much of the work presented makes use of *one* Nb clone, E_142, additional experiments focused on this NPF peptide-harboring nanobody. First, in side-by-side comparison of Nb E_142 binding in biochemical assays, no interaction with the human intersectin 1 (ITSN1) EH 1-2 array (amino acids 1-312) is seen. This data is now presented in Figure 3B, and is discussed in the text as follows:

“The EH domains of intersectin 1 similarly bind to NPF motifs [29], and the two N-terminal *Hs* intersectin 1 EH domains are about 43% identical to one another while the EH1-2 array (residues 1-312) [62] is roughly 39% identical to EPS15 EH1-3 (residues 1-314) and 36% identical to EPS15R EH1R-3R (residues 1-365). Yet in the same binding assay, Nb E_142 does not interact appreciably with GST-intersectin 1 (1-312) although the unmistakable Nb binding to both GST-EPS15 (1-109/EH1) and GST-EPS15 (1-217/EH1+2) is apparent (Figure 3B). This indicates importantly that additional non-NPF side chains must contribute to the selective binding of Nb E_142 to the EPS15/R EH domains, and likely accounts for the markedly higher apparent affinity of this Nb over the native NPF ligand for EPS15/R EH domains. Also, since none of the anti-EPS15 Nbs display acidic residues trailing the NPF in CDR3, it makes it improbable that Nb E_142 binds to other proteins with C-terminal EH domains, like EHD1 [63]”

As the reviewers stated in the critique, this knowledge is important in the overall understanding of the specificity and selectivity of these nanobodies that display orthosteric competition with endogenous NPF motif bearing endogenous EPS15/R partner proteins. I thank the reviewers for prodding me to consider this aspect more carefully.

Second, to obtain an estimate of the binding affinity of Nb E_142 for a single or triple EH domain array fused to GST, quantitative pull-down assays are now presented in Figure 3C. The accompanying narrative text is found in subsection “Characterization of a function-blocking Nb” of the revision:

“Biochemical titrations better gauge the affinity of Nb E_142 for EPS15 EH domains. Using 2 μM Nb E_142 and immobilized GST-EPS15 (1-109/EH1) varied from 0.5 to 20 μM, saturation occurs at around 10 μM and half-maximal binding occurs at ~3 μM (Figure 3C-D). With the larger GST-EPS15 (1-314/EH1-3) fusion, half-maximal binding is also seen at ~2 μM (Figure 3D and Figure 3—figure supplement 1). That the apparent *K*_D_ for Nb E_142 is roughly similar for the two fusion proteins indicates that binding to each EH domain is independent and of similar affinity. Thus Nb E_142 binds to the EH domain with low μM affinity at physiological ionic strength; these results demonstrate that the *K*_D_ of Nb E_142 for an EH domain is considerably better than a single endogenous NPF peptide [29, 59].”

These experiments show that the Nb binds to an EH domain roughly 50 times better than a native NPF peptide (*K*_D_ ~50-100 μM) and about fivefold more tightly than a 'stapled' conformationally restricted NPF motif. Together with the failure to detect Nb E_142 binding to GST-intersectin 1l (1-312), the data indicate the new anti-Eps15/R nanobodies must contact the EH domain with side chains in addition to the NPF ligand to achieve the observed specificity and apparent affinity.

Third, comparison of the mNG-NbE_142 fusion, which binds only weakly to intracellular EPS15/R at clathrin-coated structures in transiently transfected HeLa cells, with a new tandem mNG-Nb E_142 x 3 protein is shown in Figure S2. An avidity effect of duplicating the Nb pseudoligand is clearly apparent, with clathrin-coated structures now labeled by the mNG-E_142 x 3 reporter, even at very low levels of transient expression. The added results are discussed in subsection “Nb E_142 interacts with EPS15 intracellularly” of the revision:

“The concept of avidity-dependent, regionalized NPF occupancy of the EPS15/R EH domains gathered at incipient clathrin-coated structures is supported experimentally by the intracellular distribution and positioning of a transiently transfected triple Nb E_142 array. In HeLa cells, compared with the single Nb E_142 fusion, a larger mNG-Nb E_142 × 3 protein construct (Figure 4—figure supplement 1) now labels punctate AP-2-positive structures at very low expression levels. Given that both the ectopic mNG-Nb E_142 and the mNG-Nb E_142 × 3 proteins each contain only a single fluorescent mNG domain, the increased efficiency of EPS15/R labelling by the polyvalent reporter (Figure 4—figure supplement 1) must signify better apparent affinity for the massed EH domains at the sites of clathrin-coat assembly. This indicates that through avidity-based phenomena, artificially but flexibly grouped NPF-harboring Nbs can compete in vivo with the deployment multiple NPF-motifs by endogenous EH domain partner proteins.”

These findings are interpreted to support the focal competition of monomeric NbE_142 by endogenous NPF bearing EH domain interaction partners despite the higher apparent affinity of the nanobody now reported in Figure 3C-D.

Again, I am grateful to the reviewers for encouraging me to consider these important aspects more critically.

2) The work would benefit greatly from at least some quantification of the images (Image J or cellprofiler) – for example, the amount of internalized transferrin can be quantified, and the amount of AP2 or other protein per square micron for example, can be quantified. Where a nanobody has an effect, can one correlate extent of phenotype as a function of level of nanobody? This is not meant to be onerous but should really not be too hard with available image software.

In the revised manuscript, I have attempted to support some of the cell-based findings presented as representative immunofluorescence images with quantitation, as requested by the reviewers.

In my honest opinion, any visual experimental data I include in a manuscript needs to be compelling and thoroughly reproducible without any supporting statistical quantitation; I view this personally as a defense of responsibility in data presentation and reporting. It was my judgement that this was the case for the results that I included in the original manuscript submission. Furthermore, it is not as if image quantitation techniques using software algorithms are not subject to possible operator bias, over-correction, manipulation, or introduction of errors and/or artifacts.

Nevertheless, as suggested, I have now included quantitation of three different key data sets that I judge to be vital to support the overall claims and conclusions of the manuscript:

1) The unimpeded internalization of transferrin with sequestration of endogenous EPS15 and EPS15R by Nb E_142 in the parental HeLa SS6 cell background,

2) The defective internalization of the transferrin endocytic tracer in genome-edited HeLa clone #40 cells by luring endogenous EPS15 to mitochondria with Nb E_142, and

3) The shift in clathrin-coated structure size/density in HeLa clone #40 cells expressing both Akap1-Nb E_142-iRFP and GFP-FCHO1 (1-609).

Thus, I now present statistical analysis (Figure 5—figure supplement 1) of transferrin uptake experiments in HeLa SS6 cell populations transiently transfected with the GFP-FYVE_2_-Nb E_142 plasmid to judge numerically the impact of intracellular sequestration of EPS15. Similarly, quantitation of fluorescent transferrin uptake in the triple-null HeLa clone #40 cells in the modified two-pulse uptake assay is now provided in Figure 9—figure supplement 1. Finally, computation of the change in size and number of AP-2-positive clathrin-coated structures in HeLa clone #40 cells transfected with the mitochondria-targeted Nb E_142 and/or the FCHO1 (1-609) plasmid is presented in Figure 11—figure supplement 1.

Relevant additions to the main body of the narrative text describing the quantitation have been added as required, and a description of the quantitation procedures used added to the Material and methods section.